# Towards Reliable Evaluation and Fast Training of Robust Semantic Segmentation Models

## Abstract

Adversarial robustness has been studied extensively in image classification, especially for the $\ell_\infty$-threat model, but significantly less so for related tasks such as object detection and semantic segmentation. Attacks on semantic segmentation models turn out to be harder than for image classification. We propose novel attacks and motivated by their complementary properties, we put them into an attack ensemble called SEA. We use SEA to show that existing attacks can severely overestimate the robustness of semantic segmentation models. Perhaps surprisingly, existing attempts of adversarial training for semantic segmentation turn out to yield only weakly robust models or are even completely non-robust. We investigate why previous adaptations of adversarial training to semantic segmentation failed and identify insufficient training time and number of attack steps as key elements. In turn we show how recently proposed robust IMAGENET backbones can be used to obtain adversarially robust semantic segmentation models with up to six times less training time for PASCAL-VOC and the more challenging ADE20K.

## 1 Introduction

The vulnerability of neural networks to adversarial perturbations, that is small changes of the input can drastically modify the output of the models, is now well-known (Biggio et al., 2013; Szegedy et al., 2014; Grosse et al., 2016; Jin et al., 2019) and has been extensively studied, in particular for image classification. A large amount of work has gone into developing adversarial attacks in various threat models, including $\ell_p$-bounded perturbations (Carlini and Wagner, 2017; Chen et al., 2018; Rony et al., 2019), sparse attacks (Brown et al., 2017; Croce et al., 2022), and those defined by perceptual metrics (Wong et al., 2019; Laidlaw et al., 2021). At the same time, evaluating the adversarial robustness in semantic segmentation, undoubtedly an important vision task, has received much less attention. While a few early works (Xie et al., 2017; Metzen et al., 2017; Arnab et al., 2018) have proposed methods to generate adversarial attacks in different threat models, Gu et al. (2022); Agnihotri and Keuper (2023) have recently shown that even for $\ell_\infty$-bounded attacks, significant improvements are possible over the standard PGD attack (Madry et al., 2018) based on the sum of pixel-wise cross-entropy losses. In fact, the key difference of semantic segmentation to image classification is that for the former one has to flip the predictions of all pixels, not just the prediction for the image, which is a much harder optimization problem. This could explain why not many works (Xu et al., 2021; Gu et al., 2022) have produced robust semantic segmentation models via variants of adversarial training (Madry et al., 2018). In this work, we address both the evaluation and training of adversarially robust semantic segmentation models, as detailed in the following.

**Challenges of adversarial attacks in semantic segmentation.** First, we analyze why the cross-entropy loss is not a suitable objective for generating strong adversarial attacks against semantic segmentation models. To address these issues, we propose novel loss functions leading to different weightings of pixels (Sec. 2.2). Our losses have complementary properties and, interestingly, perform best at different level of robustness. In particular, we observe that taking the worst-case over the attacks of our different losses yields large improvements for the robustness evaluation. Finally, we study several improvements for the PGD-attack (Sec. 2.3) which boost performance.

**Strong evaluation of robustness with SEA.** In Sec. 2.4, we develop SEA (Semantic Ensemble Attack), a reliable robustness evaluation for semantic segmentation with the $\ell_\infty$-threat model. Inspired by the well-known AutoAttack (Croce and Hein, 2020), we build an ensemble of complementary

Table 1: **Comparison of SOTA attacks to ensemble attack SEA.** We compare the SOTA attacks SegPGD (Gu et al., 2022) and CosPGD (Agnihotri and Keuper, 2023) to our best novel single attack using the Mask-CE loss as objective and our ensemble attack SEA for the DDC-AT model (Xu et al., 2021) trained for $\epsilon_\infty = 8/255$ (top) and our PIR-AT model trained for $\epsilon_\infty = 4/255$ (bottom) on PASCAL-VOC. SEA reveals significant overestimation of robustness of existing SOTA attacks. The DDC-AT model is completely non-robust, whereas our PIR-AT yields robust segmentation results.

| model | $\epsilon_\infty$ | Previous SOTA attacks | | | | Our work | | | |
|---|---|---|---|---|---|---|---|---|---|
| | | CosPGD (300 iter.) | | SegPGD (300 iter.) | | Mask-CE (300 iter.) | | SEA (4 x 300 iter.) | |
| | | ACC | mIoU | ACC | mIoU | ACC | mIoU | ACC | mIoU |
| DDC-AT | 2/255 | 7.1 | 3.9 | 3.9 | 2.2 | 0.4 | 0.2 | **0.3** | 0.2 |
| | 4/255 | 6.5 | 3.5 | 2.6 | 1.6 | 0.2 | 0.1 | **0.0** | 0.0 |
| | 8/255 | 4.5 | 3.2 | 2.1 | 1.3 | 0.0 | 0.0 | **0.0** | 0.0 |
| PIR-AT (ours) | 4/255 | 89.2 | 65.6 | 89.1 | 65.5 | 89.0 | 65.1 | **88.3** | 63.8 |
| | 8/255 | 78.4 | 47.9 | 75.3 | 41.8 | 73.5 | 39.9 | **71.2** | 37.0 |
| | 12/255 | 57.1 | 27.7 | 44.9 | 15.7 | 31.7 | 10.1 | **27.4** | 8.1 |

attacks based on the previously introduced loss functions and algorithms, specifically designed for flipping the decisions of *all* pixels simultaneously. In the experiments, we show that recent SOTA attacks, SegPGD (Gu et al., 2022) and CosPGD (Agnihotri and Keuper, 2023), may significantly overestimate the robustness of semantic segmentation models. Table 1 illustrates that for both the DDC-AT model of Xu et al. (2021) and our robust model (see below), our SEA (and even one our attacks which is part of SEA) consistently outperforms the existing attacks partially by large margin.

**Robust semantic segmentation models with PIR-AT.** We argue that adversarial training (AT) is much harder for semantic segmentation than for image classification due to the much more difficult attack problem. In fact, to obtain satisfactory robustness with AT we had to increase the number of epochs compared to clean training and use many attack steps. However, this yields a high computational cost, that is prohibitive for scaling up to large architectures. We reduce drastically the cost by leveraging recent progress in robust IMAGENET classifiers (Debenedetti et al., 2023; Singh et al., 2023; Liu et al., 2023). In fact, we introduce Pre-trained IMAGENET Robust AT (PIR-AT), where we initialize the backbone of the segmentation model with a $\ell_\infty$-robust IMAGENET classifier. This allows us to i) reduce the cost of AT, and ii) achieve SOTA robustness on both PASCAL-VOC (Everingham et al., 2010) and ADE20K (Zhou et al., 2019) (for which no previous robust models are available). Table 1 shows that our approach, PIR-AT (see Sec. 3 for details), attains 71.2% robust average pixel accuracy at attack radius $\epsilon_\infty = 8/255$ compared to 0.0% of DDC-AT. Since robust IMAGENET models of most SOTA architectures are now available, e.g. in RobustBench (Croce et al., 2021), we apply our PIR-AT scheme to several segmentation architectures like PSPNet (Zhao et al., 2017), UPerNet (Xiao et al., 2018b) and Segmenter (Strudel et al., 2021), with consistent improvements over AT.

## 2 ADVERSARIAL ATTACKS FOR SEMANTIC SEGMENTATION

The main challenge of adversarial attacks in semantic segmentation compared to image classification is that, for each image, they have to flip not only *one* decision but the decision for *as many pixels as possible* ($\geq 10^5$ for PASCAL-VOC and ADE20K) and that pixels cannot be optimized independently. We first discuss the general setup, and introduce existing and novel loss functions specifically targeting this more challenging problem. Then, we discuss optimization schemes to further boost performance. Finally, we present our attack ensemble SEA for reliable evaluation of adversarial robustness which combines the task-specific losses and optimization algorithm to achieve SOTA attacks.

### 2.1 SETUP

The goal of semantic segmentation consists in classifying each pixel of a given image into the $K$ available classes (corresponding to different objects or background). We denote by, $f : \mathbb{R}^{w \times h \times c} \longrightarrow$

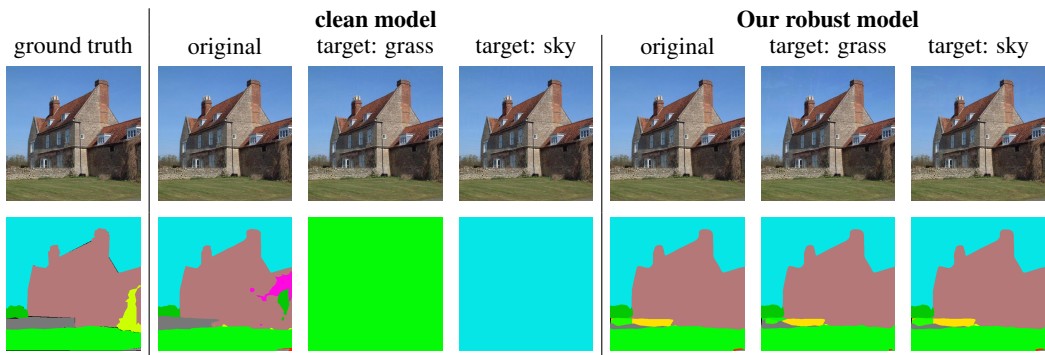

|  | | clean model | | | Our robust model | |
| ground truth | original | target: grass | target: sky | original | target: grass | target: sky |

Figure 1: **Effect of adversarial attacks on semantic segmentation models.** For a validation image of ADE20K (first column, with ground truth mask), we show the image perturbed by targeted $\ell_\infty$-attacks ($\epsilon_\infty = 2/255$, target class "grass" or "sky"), and the predicted segmentation. For a clean model the attack completely alters the segmentation, while our robust model (UPerNet +ConvNeXt-T trained with 5-step PIR-AT for 128 epochs) is minimally affected. For illustration, we use targeted attacks, and not untargeted ones as in the rest of the paper. More illustrations in App. D.

$\mathbb{R}^{w \times h \times K}$, a segmentation model which for an image $x$ of size $w \times h$ (and $c$ color channels) returns $u = f(x)$, where $u_{ij} \in \mathbb{R}^K$ contains the logit of each of the $K$ classes for the pixel $x_{ij}$. The class predicted by $f$ for $x_{ij}$ is given by $m_{ij} = \arg\max_{k=1,\dots,K} u_{ijk}$, and $m \in \mathbb{R}^{w \times h}$ is the segmentation map of $x$. Given the ground truth map $y \in \mathbb{R}^{w \times h}$, the average pixel accuracy of $f$ for $x$ is $\frac{1}{w \cdot h} \sum_{i,j} \mathbb{I}(m_{ij} = y_{ij})$. In the following, we use index $a$ for the pixel $(i, j)$ to simplify the notation. The goal of an adversarial attack on $f$ is to change the input $x$ such that as many pixels as possible are mis-classified. This can be formalized as the optimization problem:

$$\max_{\boldsymbol{\delta}} \sum_a \mathcal{L}(f(\boldsymbol{x} + \boldsymbol{\delta})_a, \boldsymbol{y}_a) \quad \text{s. th.} \quad \|\boldsymbol{\delta}\|_\infty \leq \epsilon, \quad \boldsymbol{x} + \boldsymbol{\delta} \in [0, 1]^{wh \times c}, \tag{1}$$

where we use an $\ell_\infty$-threat model for the perturbations $\delta$, $x + \delta$ is restricted to be an image, and $\mathcal{L} : \mathbb{R}^K \times \mathbb{R} \longrightarrow \mathbb{R}$ is a differentiable loss whose maximization induces mis-classification. This can then be (approximately) solved by techniques for constrained optimization such as projected gradient descent (PGD) as suggested for image classification in Madry et al. (2018).

**Background pixels.** In semantic segmentation, it is common to exclude pixels that belong to the background class when training or for computing the test performance. However, it is unrealistic for an attacker to modify only non-background pixels. Thus semantic segmentation models must be robust for all pixels regardless of how they are classified or what their ground-truth label is. Therefore, we train all our models with an additional background class. This has little impact on segmentation performance, but allows a more realistic definition of adversarial robustness, see App. C.3.

**Metrics.** In semantic segmentation it is common to use the Intersection over Union (IoU) as performance metric, averaged over classes (mIOU). The mIOU is typically computed across all the images in the test set. This means that an attacker would need to know the entire test set in order to minimize mIOU, which is impossible in practice. Therefore, similar to previous work (Agnihotri and Keuper, 2023), we use the classification accuracy averaged over pixels (ACC) as target for the attacker but in addition we report mIOU. For the worst-case over multiple attacks, we select the perturbation that gives the lowest average accuracy for each image and use it to compute the mIOU.

## 2.2 OBJECTIVE FUNCTIONS FOR ADVERSARIAL ATTACKS ON SEMANTIC SEGMENTATION

In the following, we discuss loss functions that have been used by existing attacks on semantic segmentation. We then introduce novel complementary losses which are the basis of SEA, our ensemble attack for semantic segmentation. The main challenge of the attack is to flip the decisions of most pixels of an image, ideally with only a few iterations. To tackle this problem it is therefore illustrative to consider the gradient of the total loss with respect to the input. We denote by $u \in$

$\mathbb{R}^{wh \times K}$ the logit output for the full image and by $u_{ak}$ the logit of class $k \in \{1, \dots, K\}$ for pixel $a$.

$$\nabla_{\boldsymbol{x}} \sum_a \mathcal{L}(u_a, y_a) = \sum_a \sum_{k=1}^{K} \frac{\partial \mathcal{L}}{\partial u_{ak}}(u_a, y_a) \nabla_{\boldsymbol{x}} u_{ak} \tag{2}$$

The term $\sum_{k=1}^{K} \frac{\partial \mathcal{L}}{\partial u_{ak}}(u_a, y_a)$ can be interpreted as the influence of pixel $a$. The main problem is that successfully attacked pixels (pixels wrongly classified) typically have non-zero gradients and the gradients of standard losses like cross-entropy loss have the largest magnitude for mis-classifications. Thus misclassified pixels have the strongest influence on the change of the input and in the worst case prevent that the decisions of still correctly classified pixels are flipped. The losses we discuss in the following have all their own strategy how to cope with this problem and all yield either implicitly or explicitly a different weighting of the contributions of each pixel.

Some of the losses are easier described in terms of the predicted probability distribution $\boldsymbol{p} \in \mathbb{R}^K$ via the softmax function: $\boldsymbol{p}_r = e^{\boldsymbol{u}_r} / \sum_{k=1}^{K} e^{\boldsymbol{u}_k}$, $k = 1, \dots, K$. In this case one has to understand $\boldsymbol{p}$ as a function of $\boldsymbol{u}$. Moreover, we omit the pixel-index $a$ in the following for easier presentation.

### 2.2.1 LOSSES USED IN PREVIOUS WORK FOR ATTACKS ON SEMANTIC SEGMENTATION

**Cross-entropy (CE):** the most common choice as objective function in PGD based attacks for image classification is the cross-entropy loss , i.e. $\mathcal{L}_{\text{CE}}(\boldsymbol{u}, y) = -\log \boldsymbol{p}_y = -\boldsymbol{u}_y + \log \left( \sum_{j=1}^{K} e^{\boldsymbol{u}_j} \right)$. Maximizing CE-loss is equivalent to minimizing the predicted probability of the correct class. The cross-entropy loss is unbounded, which is problematic for semantic segmentation as misclassified pixels are still optimized instead of focusing on correctly classified pixels. It holds

$$\frac{\partial \mathcal{L}_{\text{CE}}(\boldsymbol{p}, \boldsymbol{e}_y)}{\partial \boldsymbol{u}_k} = -\delta_{yk} + \boldsymbol{p}_k \quad \text{and} \quad \frac{K}{K-1}(1 - \boldsymbol{p}_y) \le \|\nabla_{\boldsymbol{u}} \mathcal{L}_{\text{CE}}\|_2^2 \le (1 - \boldsymbol{p}_y)^2 + 1 - \boldsymbol{p}_y.$$

The bounds on gradient norm are are monotonically increasing as $\boldsymbol{p}_y \to 0$ (see App. A). Therefore maximally misclassified pixels have the strongest influence in Eq. (2) for the CE-loss which explains why it does not work well as an objective for attacks on semantic segmentation (see Table 2).

**Balanced cross-entropy:** Gu et al. (2022) propose to balance in their SegPGD-attack the importance of the cross-entropy loss of correctly and wrongly classified pixels over iterations. In particular, at iteration $t = 1, \dots, T$, they use, with $\lambda(t) = (t-1)/(2T)$,

$$\mathcal{L}_{\text{Bal-CE}}(\boldsymbol{u}, y) = \left( (1 - \lambda(t)) \cdot \mathbb{I}(\underset{j=1,\dots,K}{\arg\max} \, \boldsymbol{u}_j = y) + \lambda(t) \cdot \mathbb{I}(\underset{j=1,\dots,K}{\arg\max} \, \boldsymbol{u}_j \ne y) \right) \cdot \mathcal{L}_{\text{CE}}(\boldsymbol{u}, y).$$

In this way the attack first focuses only on the correctly classified pixels and then progressively increases the weight of misclassified pixels until at iteration $T$ one optimizes the CE-loss. While this resolves some issues of the optimization of the standard CE-loss, one problem is that $\lambda$ is the same for all pixels and does not take into account the current state of the attack.

**Weighted cross-entropy:** Agnihotri and Keuper (2023) propose for their CosPGD-attack to weigh the importance of the pixels via the cosine similarity between the prediction vector $\boldsymbol{u}$ (after applying the sigmoid function $\sigma(t) = 1/(1 + e^{-t})$) and the one-hot encoding $\boldsymbol{e}_y$ of the ground truth class as $\mathcal{L}_{\text{CosSim-CE}}(\boldsymbol{u}, y) = \langle \sigma(\boldsymbol{u}), \boldsymbol{e}_y \rangle / (\|\sigma(\boldsymbol{u})\|_2 \|\boldsymbol{e}_y\|_2) \cdot \mathcal{L}_{\text{CE}}(\boldsymbol{u}, y)$. In the attack they use as "gradient"[1]:

$$\text{“}\nabla_{\boldsymbol{u}} \mathcal{L}_{\text{CosSim-CE}}(\boldsymbol{u}, y)\text{”} = \frac{\langle \sigma(\boldsymbol{u}), \boldsymbol{e}_y \rangle}{\|\sigma(\boldsymbol{u})\|_2 \|\boldsymbol{e}_y\|_2} \cdot \nabla_{\boldsymbol{u}} \mathcal{L}_{\text{CE}}(\boldsymbol{u}, y) = \sigma(\boldsymbol{u}_y) / \|\sigma(\boldsymbol{u})\|_2 \cdot \nabla_{\boldsymbol{u}} \mathcal{L}_{\text{CE}}(\boldsymbol{u}, y).$$

It holds $\sigma(\boldsymbol{u}_y) / \|\sigma(\boldsymbol{u})\|_2 \to 0$ iff $\boldsymbol{u}_y \to -\infty$ and $y = \arg\min_t \boldsymbol{u}_t$ and thus it down-weights misclassified pixels in (2). A problem is that the employed "gradient" is not the gradient of their loss.

### 2.2.2 NOVEL LOSSES FOR ATTACKS ON SEMANTIC SEGMENTATION

To counter the limitations of the existing losses, we propose losses with more explicit forms of weights: these show quite complementary properties when used in an attack, while having better or similar attack performance than the previous variants of the cross-entropy loss.

---

[1]communication with the authors as no code is available

Table 2: **Attack performance of different losses.** We optimize each loss of Sec. 2.2 with APGD for 100 iterations to attack a clean and a robust model trained on PASCAL-VOC. We report average pixel accuracy and mIOU (clean performance is next to the model name). In most cases our novel losses achieve the best performance. The best performing loss depends on $\epsilon_\infty$. The image-wise worst-case over all losses is much lower than each individual one which motivates our ensemble attack SEA.

| $\epsilon_\infty$ | losses used in prior works | | | | | | proposed losses | | | | | | Worst case over losses | |
|---|---|---|---|---|---|---|---|---|---|---|---|---|---|---|
| | $\mathcal{L}_{\text{CE}}$ | | $\mathcal{L}_{\text{Bal-CE}}$ | | $\mathcal{L}_{\text{CosSim-CE}}$ | | $\mathcal{L}_{\text{JS}}$ | | $\mathcal{L}_{\text{Mask-CE}}$ | | $\mathcal{L}_{\text{Mask-Sph.}}$ | | | |
| **UPerNet with ConvNeXt-T backbone (clean training)** | | | | | | (93.4 | 77.2) | | | | | | | |
| 0.25/255 | 77.3 | 48.3 | 74.0 | 43.7 | 76.6 | 48.0 | 73.9 | 44.3 | **73.2** | **42.7** | 75.8 | 44.4 | 71.1 | 39.9 |
| 0.5/255 | 49.3 | 25.1 | 42.3 | 18.5 | 46.9 | 24.0 | 39.4 | 18.3 | **36.9** | 14.9 | 37.5 | **14.5** | 32.5 | 12.1 |
| 1/255 | 21.2 | 9.9 | 13.9 | 4.8 | 17.2 | 8.1 | 9.1 | 4.0 | 7.9 | 2.2 | **6.6** | **1.6** | 5.3 | 1.2 |
| 2/255 | 7.4 | 4.0 | 2.9 | 1.5 | 3.4 | 2.3 | 0.5 | 0.4 | 0.3 | 0.2 | **0.1** | **0.1** | 0.1 | 0.0 |
| **Robust UPerNet with ConvNeXt-T backbone (PIR-AT (ours) - training for $\epsilon_\infty = 4/255$)** | | | | | | (92.7 | 75.9) | | | | | | | |
| 4/255 | 88.9 | 65.7 | 88.7 | **64.8** | 88.9 | 65.4 | **88.4** | **64.8** | 88.9 | 65.6 | 90.4 | 69.7 | 88.3 | 64.4 |
| 8/255 | 78.9 | 48.9 | **74.2** | **41.3** | 77.8 | 47.3 | 75.3 | 43.5 | 74.6 | 41.8 | 80.3 | 49.6 | 72.3 | 38.4 |
| 12/255 | 59.9 | 28.9 | 43.3 | 14.9 | 56.6 | 26.4 | 45.1 | 18.6 | **38.8** | 13.2 | 38.9 | **12.1** | 31.9 | 8.4 |
| 16/255 | 41.5 | 18.1 | 20.7 | 5.7 | 34.0 | 15.3 | 19.1 | 7.4 | 12.9 | 3.4 | **8.4** | **2.0** | 6.4 | 1.1 |

**Jensen-Shannon (JS) divergence:** the main problem of the cross-entropy loss is that the norm of the gradient is increasing as $\boldsymbol{p}_y \to 0$, that is the more the pixel is mis-classified. The Jensen-Shannon divergence has a quite different behavior which makes it a much better choice as attack loss function. Given two distributions $\boldsymbol{p}$ and $\boldsymbol{q}$, the Jensen-Shannon divergence is defined as

$$D_{\text{JS}}(\boldsymbol{p} \,\|\, \boldsymbol{q}) = \left(D_{\text{KL}}(\boldsymbol{p} \,\|\, \boldsymbol{m}) + D_{\text{KL}}(\boldsymbol{q} \,\|\, \boldsymbol{m})\right)/2, \quad \text{with} \quad \boldsymbol{m} = (\boldsymbol{p} + \boldsymbol{q})/2,$$

where $D_{\text{KL}}$ indicates the Kullback–Leibler divergence. Let $\boldsymbol{p}$ to be the softmax output of the logits $\boldsymbol{u}$ and $\boldsymbol{e}_y$ the one-hot encoding of the target $y$, then we set $\mathcal{L}_{\text{JS}}(\boldsymbol{u}, y) = D_{\text{JS}}(\boldsymbol{p} \,\|\, \boldsymbol{e}_y)$. As $D_{\text{JS}}$ measures the similarity between two distributions, maximizing $\mathcal{L}_{\text{JS}}$ drives the prediction of the model away from the ground truth $\boldsymbol{e}_y$. Unlike the CE loss, the JS divergence is bounded, and thus the influence of every pixel is limited. In particular, the gradient of $\mathcal{L}_{\text{JS}}$ vanishes as $\boldsymbol{p}_y \to 0$ (see App. A for the proof)

$$\lim_{\boldsymbol{p}_y \to 0} \frac{\partial \mathcal{L}_{\text{JS}}(\boldsymbol{u}, y)}{\partial \boldsymbol{u}_k} = 0, \quad \text{for} \quad k = 1, \dots, K.$$

Thus the $\mathcal{L}_{\text{JS}}$ loss automatically down-weights contributions from mis-classified pixels and thus pixels which are still correctly classified get a higher weight in the gradient in (2).

**Masked cross-entropy:** in order to avoid over-optimizing mis-classified pixels one can apply a mask which excludes such pixels from the loss computation, that is

$$\mathcal{L}_{\text{Mask-CE}} = \mathbb{I}(\arg\max_{j=1,\dots,K} \boldsymbol{u}_j = y) \cdot \mathcal{L}_{\text{CE}}(\boldsymbol{u}, y).$$

The downside of a mask is that the loss becomes discontinuous and ignoring mis-classified pixels can lead to changes which revert back mis-classified pixels into correctly classified ones with the danger of oscillations. We note that Xie et al. (2017) used masking for a margin based loss and Metzen et al. (2017) used masking for targeted attacks, to not optimize pixels already classified into the target class with confidence higher than a fixed threshold. However, the masked CE-loss has not been thoroughly explored for $\ell_\infty$-bounded untargeted attacks and turns out to be a very strong baseline, see Table 2.

**Masked spherical loss:** Croce and Hein (2020) show that the gradient of the cross-entropy can be zero due to of numerical errors when computing the softmax function, leading to gradient masking. Losses based on logit differences (Carlini and Wagner, 2017; Croce and Hein, 2020) avoid these problems. However, our attempts with purely logit-based losses were not successful. Instead we use the logit-based spherical loss (Bickel, 2007) which is a proper multi-class loss:

$$\mathcal{L}_{\text{Mask-Sph.}}(\boldsymbol{u}, \boldsymbol{e}_y) = -\mathbb{I}(\arg\max_{j=1,\dots,K} \boldsymbol{u}_j = y) \cdot \boldsymbol{u}_y / \|\boldsymbol{u}\|_2 \,,$$

using a mask for correctly classified pixels. We discuss its gradient in App. A. The masked spherical loss is particularly effective for large radii which is due to the effect that it aims at balancing the logits of all pixels which makes it then easier to flip all decisions.

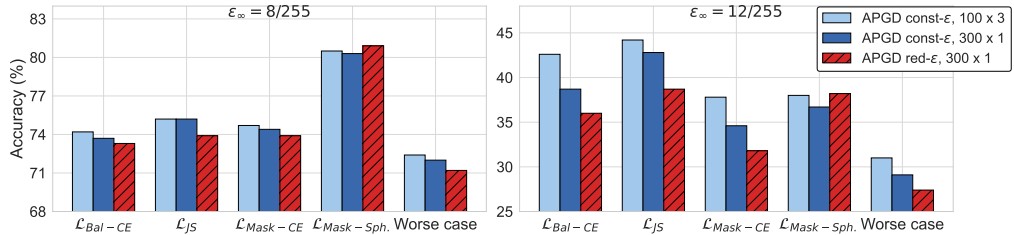

Figure 2: **Comparison of const-$\epsilon$- and red-$\epsilon$ optimization schemes.** Attack accuracy for the robust PIR-AT model from Table 1 on PASCAL-VOC, across different losses for the same iteration budget. The radius reduction (red-$\epsilon$) scheme performs best across all losses, except for $\mathcal{L}_{\text{Mask-Sph.}}$, and $\epsilon_\infty$, and even the worst-case over all losses improves. The same holds for mIOU (Fig. 3 in App. C.2).

### 2.2.3 PERFORMANCE OF DIFFERENT LOSSES

In Table 2 we compare the attack performance of all losses discussed in this section when optimized with APGD (see Sec. 2.3) for 100 iterations on PASCAL-VOC models. For the clean model, all of our proposed losses yield better results than existing losses of Gu et al. (2022); Agnihotri and Keuper (2023) for every $\epsilon_\infty$. For the robust model, our Mask-CE is comparable or better than Bal-CE for small $\epsilon_\infty$ but performs much better for $\epsilon_\infty \geq 12/255$. However, the more important observation is that the image-wise worst case over all losses yields improvements of up to 6.9% in average accuracy and 3.7% in mIOU. This motivates our ensemble attack SEA which we introduce in Sec. 2.4.

### 2.3 OPTIMIZATION TECHNIQUES FOR ADVERSARIAL ATTACKS ON SEMANTIC SEGMENTATION

A good adversarial attack should yield robust accuracies which are close to optimal but at the same time use the smallest amount of computational resources possible. In the following we discuss different measures how to optimize this trade-off.

**APGD vs PGD for Semantic Segmentation Attacks.** We use Projected gradient descent (PGD) (Madry et al., 2018) for *minimizing* an objective function $g$ in the set $S = \left\{ z \in \mathbb{R}^d : \|z - x\|_\infty \leq \epsilon, z \in [0, 1]^d \right\}$, i.e. an $\ell_\infty$-ball around a given point $x$ intersected with the image box, denoting by $P_S$ the projection onto $S$. The iterations with stepsize $\alpha^{(t)}$ are given as

$$x^{(t+1)} = P_S \left( x^{(t)} - \alpha^{(t)} \operatorname{sign}(\nabla g(x^{(t)})) \right).$$

Typically, prior work uses a fixed step size. However, fixed step sizes have been shown to be suboptimal in Croce and Hein (2020) and they propose APGD which chooses step size and restarts adaptively, with no free parameters except the total number of iterations. While APGD was designed for image classification, it can be applied to general objectives with constraint set $S$. We show in Table 7 in the Appendix that simply replacing PGD with APGD improves the performance of SegPGD (Gu et al., 2022) (loss: $\mathcal{L}_{\text{Bal-CE}}$) and CosPGD (Agnihotri and Keuper, 2023) (loss: $\mathcal{L}_{\text{CosSim-CE}}$), when attacking the robust model trained with PIR-AT from Table 1. For both attacks and almost all radii $\epsilon_\infty$, APGD yields lower average pixel accuracy and mIOU with the same cost as PGD. The improvements are large for high values of $\epsilon$. Thus we use APGD for the optimization of all losses in this paper.

**Progressive radius reduction.** In Table 2, the worst case over losses is in some cases much lower than each individual attack, suggesting that the optimization can get stuck regardless of the objective function. At the same time, increasing the radius, i.e. larger $\epsilon_\infty$, reduces robust accuracy eventually to zero. Thus the gradient information is valid and there is no gradient masking. In order to mitigate this problem and profit from the result of larger $\epsilon_\infty$, we run the attack starting with a larger radius and then use its projection onto the feasible set as starting point for the attack with a smaller radius, similar to Croce and Hein (2021) for $\ell_1$-attacks. We split the budget of iterations into three slots (with ratio $3 : 3 : 4$) with attack radii $2 \cdot \epsilon_\infty$, $1.5 \cdot \epsilon_\infty$ and $\epsilon_\infty$ respectively.

**Radius reduction vs more iterations.** To assess the effectiveness of the scheme with progressive reduction of the radius $\epsilon$ (red-$\epsilon$) described above, we compare it to the standard scheme (const-$\epsilon$) for a fixed budget of 300 iterations. For const-$\epsilon$, we do either 300 iterations or 3 random restarts with 100 iterations each, and 300 iterations for (red-$\epsilon$). In Fig. 2 we show the robust accuracy achieved by the three attacks with different losses, for $\epsilon_\infty \in \{8/255, 12/255\}$, on our robust PIR-AT-model

Table 3: **Evaluation of our robust models with SEA on PASCAL-VOC and ADE20K.** For each model and choice of $\epsilon_\infty$ we report the training details (clean/robust initialization for backbone, number of attack steps and training epochs) and robust average pixel accuracy (white background) and mIoU (grey background) evaluated with SEA. * indicates results reported in Gu et al. (2022) evaluated with 100 iterations of SegPGD (models are not available) which is much weaker than SEA.

| Training | Init. | Steps | Architecture | 0 | | 4/255 | | 8/255 | | 12/255 | |
|---|---|---|---|---|---|---|---|---|---|---|---|
| **PASCAL-VOC: 50 epochs** | | | | | | | | | | | |
| DDC-AT (Xu et al., 2021) | clean | 3 | PSPNet + RN-50 | 95.1 | 75.9 | 0.0 | 0.0 | 0.0 | 0.0 | 0.0 | 0.0 |
| AT (Xu et al., 2021) | clean | 3 | PSPNet + RN-50 | 94.0 | 74.1 | 0.0 | 0.0 | 0.0 | 0.0 | 0.0 | 0.0 |
| SegPGD-AT (Gu et al., 2022) | clean | 7 | PSPNet + RN-50 | – | 74.5 | – | – | – | 17.0* | – | – |
| PIR-AT (ours) | robust | 5 | PSPNet + RN-50 | 90.6 | 68.9 | **81.8** | 49.0 | **51.3** | 15.3 | **13.4** | **2.6** |
| AT (ours) | clean | 5 | UPerNet + CN-T | 91.9 | 73.1 | 86.2 | 59.2 | 64.6 | 28.3 | 20.7 | 4.9 |
| PIR-AT (ours) | robust | 5 | UPerNet + CN-T | **92.7** | **75.2** | **88.3** | **63.8** | **71.2** | **37.0** | **27.4** | **8.1** |
| AT (ours) | clean | 5 | UPerNet + CN-S | 92.4 | 74.6 | 86.8 | 63.2 | 68.4 | 33.8 | 24.2 | 6.1 |
| PIR-AT (ours) | robust | 5 | UPerNet + CN-S | **93.1** | **76.6** | **89.2** | **66.2** | **70.8** | **38.0** | **27.0** | **8.6** |
| **ADE20K: 128 epochs** | | | | | | | | | | | |
| AT (ours) | clean | 5 | UPerNet + CN-T | 68.0 | 26.1 | 52.4 | 14.0 | 24.7 | 4.7 | 2.4 | 0.3 |
| PIR-AT (ours) | robust | 5 | UPerNet + CN-T | **70.5** | **31.7** | **55.6** | **18.6** | **26.4** | **6.7** | **3.3** | **0.8** |
| AT (ours) | clean | 5 | Segmenter + ViT-S | 67.7 | 26.8 | 48.4 | 12.6 | 25.0 | 4.7 | 4.5 | 0.8 |
| PIR-AT (ours) | robust | 5 | Segmenter + ViT-S | **69.1** | **28.7** | **54.5** | **16.1** | **32.8** | **7.1** | **8.6** | **1.8** |

from Table 1. Our progressive reduction scheme red-$\epsilon$ APGD yields the best results (lowest accuracy) for almost every case, with large improvements especially at $\epsilon_\infty = 12/255$. This suggests that this scheme is better suited for generating stronger attacks on semantic segmentation models than common options used in image classification like more iterations or random restarts.

## 2.4 SEGMENTATION ENSEMBLE ATTACK (SEA)

Based on the findings about the complementary properties of loss functions for different radii $\epsilon_\infty$ and the improved optimization by the progressive radius reduction, we propose the Segmentation Ensemble Attack (SEA) as an evaluation protocol for segmentation models. SEA consists of one run of 300 iterations with red-$\epsilon$ APGD for each of the four best losses found above, namely $\mathcal{L}_{\text{Mask-CE}}$, $\mathcal{L}_{\text{Bal-CE}}$, $\mathcal{L}_{\text{JS}}$ and $\mathcal{L}_{\text{Mask-Sph}}$. We then take the image-wise worst-case (across the 4 losses) based on ACC and report the mean. Our choice is motivated by the fact that the worst-case over these four losses results in maximum $0.1\%$ higher robust average accuracy or mIoU than using all six losses. Thus the two left-out losses, $\mathcal{L}_{\text{CE}}$ and $\mathcal{L}_{\text{CosSim-CE}}$, do not add any benefit (see Table 8). We analyze SEA in detail in App. C.2, e.g. using only three losses or just 100 iterations leads to an overestimation of robust accuracy of up to $3.0\%$, whereas a higher budget of 500 iterations does not help.

**Comparison to prior work.** We compare our attacks to the CosPGD and SegPGD attack in Table 1, where we evaluate all attacks at various radii $\epsilon_\infty$ for a DDC-AT model and our robust PIR-AT model trained on PASCAL-VOC. The mask-CE loss (one of the components of SEA) optimized with red-$\epsilon$ APGD already outperforms, with the same computational cost of 300 iterations, the prior attacks for every model and radius. SEA reduces the balanced accuracy compared to CosPGD by up to 6.8% for the clean and up to 29.7% for the robust model and compared to SegPGD by up to $4.5\%$ for the clean and $4.1\%$ for the robust model. This shows that SEA achieves significant gains compared to previous work and enables a reliable robustness evaluation for semantic segmentation.

## 3 ADVERSARIALLY ROBUST SEGMENTATION MODELS

Adversarial training (AT) (Madry et al., 2018) and its variants are the established technique to get adversarially robust image classifiers. One major drawback is the significantly longer training time

due to the adversarial attack steps ($k$ steps imply a factor of $k+1$ higher cost). This can be prohibitive for large backbones and decoders in semantic segmentation models. As a remedy we propose Pre-trained ImageNet Robust Models AT (PIR-AT), which can reduce the training time by a factor of 4-6 while improving the SOTA of robust semantic segmentation models.

**Experimental setup.** In this section, we use PGD for adversarial training with the cross-entropy loss and an $\ell_\infty$-threat model of radius $\epsilon_\infty = 4/255$ (as used by robust classifiers on IMAGENET). We also tried AT with the losses from Sec. 2.2 but got no improvements over AT with cross-entropy loss. For training configuration, we mostly follow standard practices for each architecture (Zhao et al., 2017; Liu et al., 2022; Strudel et al., 2021), in particular for the number of epochs (see App. B for training and evaluation details). All robustness evaluations are done with SEA on the entire validation set.

### 3.1 PIR-AT: ROBUST MODELS VIA ROBUST INITIALIZATION

When training clean and robust semantic segmentation models it is common practice to initialize the backbone with a clean classifier pre-trained on IMAGENET. In contrast, we propose for training robust models to use $\ell_\infty$-robust IMAGENET models as initialization of the backbone (the decoder is always initialized randomly), and we name such approach Pre-trained ImageNet Robust Models AT (PIR-AT). This seemingly small change has huge impact on the final outcome of the model. We show in Table 3 a direct comparison of AT and PIR-AT using the same number of adversarial steps and training epochs across different decoders and architectures for the backbones. In all cases PIR-AT outperforms AT in clean and robust accuracy. Up to our knowledge these are also the first robust models for ADE20K. Regarding PASCAL-VOC, we note that DDC-AT (Xu et al., 2021) as well as their AT-model are completely non-robust. SegPGD-AT, trained for $\epsilon_\infty = 8/255$ and with the larger attack budget of 7 steps, is seemingly more robust but is evaluated with only 100 iterations of SegPGD which is significantly weaker than SEA (models are not available) and therefore their robustness could be overestimated. However, our UPerNet + ConvNeXt-T (CN-T) outperforms SegPGD-AT by at least 20% in mIOU at $\epsilon_\infty = 8/255$ even though it is trained for $4/255$. Interestingly, the large gains in robustness do not degrade much the clean performance, which is a typical drawback of adversarial training. Our results for ConvNeXt-S (CN-S) show that this scales to larger backbones.

### 3.2 ABLATION STUDY OF AT VS PIR-AT

In Table 4 we provide a detailed comparison of AT vs PIR-AT for different number of adversarial steps and training epochs. On PASCAL-VOC, for 2 attack steps, AT with clean initialization for 50 epochs does not lead to any robustness. This is different to image classification where 50 epochs 2-step AT are sufficient to get robust models on IMAGENET (Singh et al., 2023). In contrast 2-step PIR-AT yields a robust model and even outperforms 300 epochs of AT with clean initialization in terms of robustness at all $\epsilon_\infty$ while being 6 times faster to train. This shows the significance of the initialization. For 5 attack steps we see small improvements of robustness at $4/255$ compared to 2 steps but much larger ones at $8/255$. Again, 50 epochs of PIR-AT perform roughly the same as 300 epochs of AT with clean initialization. For ADE20K we can make similar observations, which generalize across architectures (UPerNet + ConvNeXt-T and Segmenter+ViT-S): 32 epochs of PIR-AT outperform 128 epochs of AT with clean initialization in terms of robustness (4 times faster). As robust IMAGENET classifiers are now available for different architectures and sizes, we believe that PIR-AT should become standard to train robust semantic segmentation models.

## 4 RELATED WORK

**Adversarial attacks for semantic segmentation.** $\ell_\infty$-bounded attacks on segmentation models have been first proposed by Metzen et al. (2017), which focus on targeted (universal) attacks, and Arnab et al. (2018), using FGSM (Goodfellow et al., 2015) or PGD on the cross-entropy loss. Recently, Gu et al. (2022); Agnihotri and Keuper (2023) revisited the loss used in the attack to improve the effectiveness of $\ell_\infty$-bounded attacks, and are closest in spirit to our work. Additionally, there exist a few works with attacks for other threat models, including unconstrained, universal and patch attacks (Xie et al., 2017; Cisse et al., 2017; Mopuri et al., 2018; Shen et al., 2019; Kang et al., 2020; Nesti et al., 2022; Rony et al., 2023). In particular, Rony et al. (2023) introduce an algorithm to minimize the $\ell_\infty$-norm of the perturbations such that a fixed percentage of pixels is successfully attacked. While

Table 4: **Ablation study AT vs PIR-AT.** We show the effect of varying the number of attack steps and training epochs on the robustness (measured with ACC and mIoU at various radii) of the models trained with AT and PIR-AT. Our PIR-AT achieves similar or better robustness than AT at significantly reduced computational cost for all datasets and architectures.

| Training Scheme | Init. | Steps | Ep. | 0 | | 4/255 | | 8/255 | | 12/255 | |
|---|---|---|---|---|---|---|---|---|---|---|---|
| **PASCAL-VOC, UPerNet with ConvNeXt-T backbone** | | | | | | | | | | | |
| AT (ours) | clean | 2 | 50 | 93.4 | 77.4 | 2.6 | 0.1 | 0.0 | 0.0 | 0.0 | 0.0 |
| PIR-AT (ours) | robust | 2 | 50 | 92.9 | 75.9 | 86.7 | 60.8 | 50.2 | 21.0 | 9.3 | 2.4 |
| AT (ours) | clean | 2 | 300 | 93.1 | 76.3 | 86.5 | 59.6 | 44.1 | 16.6 | 4.6 | 0.1 |
| AT (ours) | clean | 5 | 50 | 91.9 | 73.1 | 86.2 | 59.2 | 64.6 | 28.3 | 20.7 | 4.9 |
| PIR-AT (ours) | robust | 5 | 50 | 92.7 | 75.2 | 88.3 | 63.8 | 71.2 | 37.0 | 27.4 | 8.1 |
| AT (ours) | clean | 5 | 300 | 92.8 | 75.5 | 88.6 | 64.4 | 71.4 | 37.7 | 23.4 | 6.6 |
| **ADE20K, UPerNet with ConvNeXt-T backbone** | | | | | | | | | | | |
| AT (ours) | clean | 2 | 128 | 73.4 | 36.4 | 0.2 | 0.0 | 0.0 | 0.0 | 0.0 | 0.0 |
| PIR-AT (ours) | robust | 2 | 128 | 72.0 | 34.7 | 46.0 | 15.4 | 6.0 | 1.8 | 0.0 | 0.0 |
| PIR-AT (ours) | robust | 5 | 32 | 68.8 | 25.2 | 55.4 | 15.6 | 28.3 | 5.9 | 3.8 | 0.7 |
| AT (ours) | clean | 5 | 128 | 68.0 | 26.1 | 52.4 | 14.0 | 24.7 | 4.7 | 2.4 | 0.3 |
| PIR-AT (ours) | robust | 5 | 128 | 70.5 | 31.7 | 55.6 | 18.6 | 26.4 | 6.7 | 3.3 | 0.8 |
| **ADE20K, Segmenter with ViT-S backbone** | | | | | | | | | | | |
| PIR-AT (ours) | robust | 5 | 32 | 68.1 | 26.0 | 55.5 | 15.4 | 33.2 | 6.9 | 7.8 | 1.4 |
| AT (ours) | clean | 5 | 128 | 67.7 | 26.8 | 48.4 | 12.6 | 25.0 | 4.7 | 4.5 | 0.8 |
| PIR-AT (ours) | robust | 5 | 128 | 69.1 | 28.7 | 54.5 | 16.1 | 32.8 | 7.1 | 8.6 | 1.8 |

their threat model is quite different, we adapt our attacks to their threat model and do a comparison in App. C.4. In particular for robust models, we outperform them by large margin. Finally, APGD (Croce and Hein, 2020) has not been used as optimizer for attacks against segmentation models.

**Robust segmentation models.** Only a few works have developed defenses for semantic segmentation models. Xiao et al. (2018a) propose a method to detect attacks, while stating adversarial training is hard to adapt for semantic segmentation. Later, DDC-AT (Xu et al., 2021) attempts to integrate adversarial points during training exploiting additional branches to the networks. The seemingly robust DDC-AT has been shown to be non-robust using SegPGD by Gu et al. (2022) at $\epsilon_\infty = 8/255$, whereas we show with SEA (Table 1) that it is non-robust even for $\epsilon_\infty = 2/255$ where SegPGD still flags robustness. Finally, Cho et al. (2020); Kapoor et al. (2021) present defenses based on denoising the input, with either Autoencoders or Wiener filters, to remove adversarial perturbations before feeding it to clean models. These methods are only tested via attacks with limited budget, while similar techniques to protect image classifiers have proven ineffective when evaluated with adaptive attacks (Athalye et al., 2018; Tramèr et al., 2020).

## 5   CONCLUSION

We have shown that adversarial attacks on semantic segmentation models can be improved by adapting the optimization algorithms and objective functions, developing SEA, an ensemble of attacks which outperforms existing methods. Moreover, we could train segmentation models with SOTA robustness, even at reduced computational cost, by leveraging adversarially pre-trained IMAGENET classifiers. We hope that the availability of our robust models fosters research in robust semantic segmentation.

**Limitations.** We consider SEA an important step towards a strong robustness evaluation of semantic segmentation models. However, similar to AutoAttack (Croce and Hein, 2020), white-box attacks should be complemented by strong black-box attacks which we leave to future work. Moreover, several techniques, e.g. using different loss functions, unlabeled and synthetic data, adversarial weight perturbations, etc., have been shown effective to achieve more robust classifiers, and might be beneficial for segmentation too, but testing all these options is out of scope for our work.

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

CONTENTS OF THE APPENDIX

BROADER IMPACT

We propose new techniques to test the robustness of segmentation models to adversarial attacks. While we consider it important to estimate the vulnerability of existing systems, such methods might potentially be used by malicious actors. However, we also provide insights on how to obtain, at limited computational cost, models which are robust to such perturbations.

## A  PROOF OF THE PROPERTIES OF CROSS-ENTROPY, THE JENSEN-SHANNON-DIVERGENCE AND THE SPHERICAL LOSS

**Cross-entropy loss:**
The cross-entropy is given as: $\mathcal{L}_{\text{CE}}(\boldsymbol{p}, e_y) = -\log \boldsymbol{p}_y$, and has gradient

$$\frac{\partial \mathcal{L}_{\text{CE}}(\boldsymbol{u}, e_y)}{\partial u_t} = -\delta_{yt} + \boldsymbol{p}_t(u).$$

We note that

$$\|\nabla_u \mathcal{L}_{\text{CE}}(\boldsymbol{u}, e_y)\|_2^2 = \sum_{t \neq y} \boldsymbol{p}_t^2 + (1 - \boldsymbol{p}_y)^2.$$

As $0 \leq \boldsymbol{p}_t \leq 1$, it holds

$$\sum_{t \neq y} \boldsymbol{p}_t^2 \leq \sum_{t \neq y} \boldsymbol{p}_t = 1 - \boldsymbol{p}_y$$

and the point of minimal $\ell_2$-distance on the surface of the $\ell_1$-ball with radius $1 - \boldsymbol{p}_y$ has equal components and thus

$$\sum_{t \neq y} \boldsymbol{p}_t^2 \geq \frac{1 - \boldsymbol{p}_y}{K - 1},$$

which yields

$$\frac{K}{K - 1}(1 - \boldsymbol{p}_y) \leq \|\nabla_u \mathcal{L}_{\text{CE}}(\boldsymbol{u}, e_y)\|_2^2 \leq 1 - \boldsymbol{p}_y + (1 - \boldsymbol{p}_y)^2.$$

We note that both lower and upper bounds are monotonically increasing as $\boldsymbol{p}_y \to 0$.

**Jensen-Shannon divergence:**
The Jensen-Shannon-divergence between the predicted distribution $p$ and the label distribution $q$ is given by

$$D_{\text{JS}}(\boldsymbol{p} \,\|\, \boldsymbol{q}) = (D_{\text{KL}}(\boldsymbol{p} \,\|\, \boldsymbol{m}) + D_{\text{KL}}(\boldsymbol{q} \,\|\, \boldsymbol{m})) / 2, \quad \text{with} \quad \boldsymbol{m} = (\boldsymbol{p} + \boldsymbol{q}) / 2,$$

Assuming that we have a one-hot label encoding $\boldsymbol{q} = e_y$ (where $e_y$ is the $y$-th cartesian coordinate vector), one gets

$$D_{\text{JS}}(\boldsymbol{p} \,\|\, e_y) = \frac{1}{2} \log\left(\frac{2}{1 + \boldsymbol{p}_y}\right) + \frac{1}{2} \sum_{i=1}^{K} \boldsymbol{p}_i \log\left(\frac{2\boldsymbol{p}_i}{\delta_{yi} + \boldsymbol{p}_i}\right).$$

Then

$$\frac{\partial D_{\text{JS}}(\boldsymbol{p} \,\|\, e_y)}{\partial \boldsymbol{p}_r} = -\frac{1}{1 + \boldsymbol{p}_y}\delta_{yr} + \log\left(\frac{2\boldsymbol{p}_r}{\delta_{yr} + \boldsymbol{p}_r}\right) + 1 - \frac{\boldsymbol{p}_r}{\delta_{yr} + \boldsymbol{p}_r} = \begin{cases} \log\left(\frac{2\boldsymbol{p}_y}{1 + \boldsymbol{p}_y}\right) & \text{if } r = y, \\ \log(2) & \text{else}. \end{cases}$$

Given the logits $u$ we use the softmax function

$$\boldsymbol{p}_r = \frac{e^{u_r}}{\sum_{t=1}^{K} e^{u_t}}, \quad r = 1, \ldots, K,$$

to obtain the predicted probability distribution $\boldsymbol{p}$. One can compute:

$$\frac{\partial \boldsymbol{p}_r}{\partial u_t} = \delta_{rt}\boldsymbol{p}_t - \boldsymbol{p}_r\boldsymbol{p}_t \quad \implies \quad \sum_{r=1}^{K} \frac{\partial \boldsymbol{p}_r}{\partial u_t} = 0$$

Then

$$\frac{\partial D_{\mathrm{JS}}(\boldsymbol{p} \,\|\, e_y)}{\partial u_t} = \sum_{r=1}^{K} \frac{\partial D_{\mathrm{JS}}(\boldsymbol{p} \,\|\, e_y)}{\partial \boldsymbol{p}_r} \frac{\partial \boldsymbol{p}_r}{\partial u_t} = \log\left(\frac{2\boldsymbol{p}_y}{1+\boldsymbol{p}_y}\right) \frac{\partial \boldsymbol{p}_y}{\partial u_t} + \log(2) \sum_{r \neq y} \frac{\partial \boldsymbol{p}_r}{\partial u_t}$$

$$= \log\left(\frac{2\boldsymbol{p}_y}{1+\boldsymbol{p}_y}\right) \frac{\partial \boldsymbol{p}_y}{\partial u_t} - \log(2) \frac{\partial \boldsymbol{p}_y}{\partial u_t} = \log\left(\frac{\boldsymbol{p}_y}{1+\boldsymbol{p}_y}\right) [\delta_{yt}\boldsymbol{p}_y - \boldsymbol{p}_y\boldsymbol{p}_t]$$

$$= \boldsymbol{p}_y \log\left(\frac{\boldsymbol{p}_y}{1+\boldsymbol{p}_y}\right) [\delta_{yt} - \boldsymbol{p}_t]$$

Noting that $\lim_{x \to 0} x \log(x) = 0$ we get the result that: $\lim_{\boldsymbol{p}_y \to 0} \frac{\partial D_{\mathrm{JS}}(\boldsymbol{p} \,\|\, e_y)}{\partial u_t} = 0$.

**Spherical loss:**
The gradient of the spherical loss is given by:

$$\frac{\partial \mathcal{L}_{\mathrm{Sph.}}(\boldsymbol{u}, e_y)}{\partial \boldsymbol{u}_t} = \frac{1}{\|\boldsymbol{u}\|_2}\left(-\delta_{yt} + \frac{\boldsymbol{u}_y \boldsymbol{u}_t}{\|\boldsymbol{u}\|_2^2}\right),$$

The main difference to the cross-entropy loss is first the weighting factor $\frac{1}{\|\boldsymbol{u}\|_2}$ which aims at balancing the logits and second the occurence of $\boldsymbol{u}_y$ in the second term of the derivative. The latter leads to a direct weighting in terms of $\boldsymbol{u}_y$ (note that for the masked spherical loss it holds $\boldsymbol{u}_y \geq \boldsymbol{u}_t$ for all $t \neq y$) and thus leads to a larger influence of pixels which are still correctly classified.

**Discussion.** The (theoretical) discussion of benefits and weaknesses for each loss in Sec. 2.2 suggests that one main difference among losses is how they balance the weight of different pixels in the objective function. On one extreme, the plain cross-entropy maximizes the loss for all pixels independently of whether they are misclassified, and assign them the same importance. Conversely, the masked losses exclude (via the mask) the misclassified pixels from the objective function, with the danger of reverting back the successful perturbations. As middle ground, losses like the JS divergence assign a weight to each pixel based on how "confidently" they are misclassified. We conjecture that for radii where robustness is low, masked losses help focusing on the remaining pixels, and already misclassified pixels are hardly reverted since they are far from the decision boundary. Conversely, at smaller radii achieving confident misclassification is harder (since the perturbations are smaller), and most pixels are still correctly classified or misclassified but close to the decision boundary: then it becomes more important to balance all of them in the loss, hence losses like JS divergence are more effective. This hypothesis is in line with the empirical results in Table 2.

## B    EXPERIMENTAL DETAILS

We here provide additional details about both attacks and training scheme used in the experiments in the main part.

### B.1    ATTACKS FOR SEMANTIC SEGMENTATION

**Baselines.** Since Gu et al. (2022); Agnihotri and Keuper (2023) do not provide code for their methods, we re-implement both SegPGD and CosPGD following the indications in the respective papers and

Table 5: **Training and data configurations.** For all the models trained in this work, we list according to the dataset, the training and dataset configurations. Warmup epochs are scaled depending on the total number of epochs. Poly dec. is the polynomially decaying schedule, from Zhao et al. (2017). The setup stays the same across all setups of adversarial training (clean init./robust init. or 2 vs 5 step).

| | Configuration | PASCAL-VOC | | ADE20K | |
|---|---|---|---|---|---|
| | | PSPNet | UPerNet | UPerNet | Segmenter |
| DATA | Base size | 512 | 512 | 520 | 520 |
| | Crop size | 473x473 | 473x473 | 512x512 | 512x512 |
| | Random Horizontal Flip | ✓ | ✓ | ✓ | ✓ |
| | Random Gaussian Blur | ✓ | ✓ | ✓ | ✓ |
| TRAINING | Optimizer | SGD | AdamW | AdamW | SGD |
| | Base learning rate | 5e-4 | 1e-3 | 1e-3 | 2e-3 |
| | Weight decay | 0.0 | 1e-2 | 1e-2 | 1e-2 |
| | Batch size | 16x8 | 16x8 | 16x8 | 16x8 |
| | Epochs | 50/300 | 50/300 | 32/128 | 32/128 |
| | Warmup epochs | 5/30 | 5/30 | 5/20 | 5/20 |
| | Momentum | 0.9 | 0.9, 0.999 | 0.9, 0.999 | 0.9 |
| | LR schedule | poly dec. | poly dec. | poly dec. | poly dec. |
| | Warmup schedule | linear | linear | linear | linear |
| | Schedule power | 0.9 | 1.0 | 1.0 | 0.9 |
| | LR ratio (Enc:Dec) | 1:10 | ✗ | ✗ | ✗ |
| | Auxilary loss weight | 0.4 | 0.4 | 0.4 | – |

personal communication with the authors of CosPGD. In the comparison in Table 1, we use PGD with step size (2e-3, 4e-3, 5e-3, 6e-3) for radii (2/255, 4/255, 8/255, 12/255) resp. for both CosPGD and SegPGD for 300 iterations each. The step size selection was done via a small grid-search. Moreover, we select for each image the iterate with highest loss.

**APGD with masked losses.** Since APGD relies on the progression of the objective function value to e.g. select the step size, using losses which mask the mis-classified pixels might be problematic, since the loss is not necessarily monotonic. Then, in practice we only apply the mask when computing the gradient at each iteration.

## B.2 TRAINING ROBUST MODELS

In the following, we detail the employed network architectures, as well as our training procedure for the utilized datasets. All experiments are conducted in multi-GPU setting with `PyTorch` (Paszke et al., 2019) library. For adversarial training we use PGD at $\epsilon = 4/255$ and step size 0.01. While training clean or adversarially, the backbones are initialized with publicly available IMAGENET pre-trained models, source of which are listed in Table 6.

**Model architectures.** Semantic segmentation model architectures have adapted to use image classifiers in their backbone. UPerNet coupled with ConvNeXt (Liu et al., 2022) and transformer models like ViT (Dosovitskiy et al., 2021) with Segmenter (Strudel et al., 2021) achieve art segmentation results. We choose UPerNet and Segmenter architectures for our experiments with ConvNeXt and ViT as the resp. backbones. For direct comparison to existing robust segmentation works (Gu et al., 2022; Xu et al., 2021) which only train with a PSPNet (Zhao et al., 2017), we also train a PSPNet with a ResNet-50 backbone. Table 5 reports the training and data related information about the various architectures and the backbones used.

Table 6: **Source of our pre-trained backbones.** We employ the same backbone for both PASCAL-VOC and ADE20K. The robust column indicates if the backbone used is adversarially robust for IMAGENET and we also list the IMAGENET clean and robust accuracy at $\ell_\infty$-radius of $\epsilon = 4/255$.

| Architecture | Backbone | Robust | Source | IMAGENET acc. clean | $\ell_\infty$ |
|---|---|---|---|---|---|
| UPerNet | ConvNeXt-T + ConvStem | ✗ | Singh et al. (2023) | 80.9% | 0.0% |
| UPerNet | ConvNeXt-T + ConvStem | ✓ | Singh et al. (2023) | 72.7% | 49.5% |
| UPerNet | ConvNeXt-S + ConvStem | ✓ | Singh et al. (2023) | 74.1% | 52.4% |
| Segmenter | ViT-S | ✗ | Wightman (2019) | 81.2% | 0.0% |
| Segmenter | ViT-S | ✓ | Singh et al. (2023) | 69.2% | 44.4% |
| PSPNet | ResNet-50 | ✓ | Salman et al. (2020) | 64.0% | 35.0% |

**UPerNet with ConvNeXt backbone.** For both clean and robust initialization setups, we use the publically available IMAGENET-1k pre-trained weights[2] from Singh et al. (2023), which achieve art robustness for $\ell_\infty$-threat model at $\epsilon = 4/255$. They propose some architectural changes, notably replacing PatchStem with a ConvStem in their most robust ConvNeXt models, and we keep these changes intact in our UPerNet models, we always use a ConvNeXt with ConvStem in this work. We highlight that ConvNeXt-T, when adversarially trained for classification on IMAGENET, attains significantly higher robustness than ResNet-50 at a similar parameter and FLOPs count. For example, at $\epsilon_\infty = 4/255$, the ConvNeXt-T we use has 49.5% of robust accuracy, while ResNet-50 is reported to achieve around 35% (Salman et al., 2020; Bai et al., 2021). This supports choosing ConvNeXt as backbone for obtaining robust segmentation models with the UPerNet architecture. For UPerNet with the ConvNeXt backbone, we use the training setup from Liu et al. (2022), listed in Table 5. We also use the same values of 0.4 or 0.3 for stochastic depth coefficient depending on the backbone, same as the original work[3]. We do not use heavier augmentations and Layer-Decay (Bao et al., 2022) optimizer as done by Liu et al. (2022).

**Segmenter with ViT backbone.** Testing with Segmenter also enables a further comparison across model size as Segmenter with a ViT-S backbone is less than half the size (26 million parameters) of UPerNet with a ConvNeXt-T backbone (60 million parameters). We define the training setup in Table 5, which is similar to the setup used by Strudel et al. (2021). The decoder is a Mask transformer and is randomly initialized. Note that Strudel et al. (2021) predominantly use IMAGENET pre-trained classifiers at resolution of 384x384, whereas we use 224x224 resolution as no robust models at the higher resolution are available.

**PSPNet with ResNet backbone.** As prior works (Xu et al., 2021; Gu et al., 2022) use a PSPNet with a ResNet (He et al., 2016) backbone to test their robustness evaluations, we also train the same model for the PASCAL-VOC dataset. Both DDCAT (Xu et al., 2021) and SegPGD-AT (Gu et al., 2022) use a split of 50% clean and 50% adversarial inputs for training. Instead for PIR-AT with PSPNet, we just use adversarial inputs. Due to this change, and due to the fact that we initialize PIR-AT with IMAGENET pre-trained ResNet-50 (RN50), we slightly deviate from the standard training parameters (learning rate, weight decay, warmup epochs) as in the original PSPNet work Zhao et al. (2017). The detailed training setup is listed in Table 5.

**Training setup for PASCAL-VOC.** We use the augmentation setup from Hariharan et al. (2011). Our training set comprises of 8498 images and we validate on the original PASCAL-VOC validation set of 1449 images. Data and training configurations are detailed in Table 5. Adversarial training is done with either 2 or 5 steps of PGD with the cross-entropy loss. Unlike some other works in literature, we train for 21 classes (including the background class).

**Training setup for ADE20K.** We use the full standard training and validation sets from Zhou et al. (2019). Adversarial training is done with either 2 or 5 steps of PGD with the cross-entropy loss. Unlike the original work we train with 151 classes (including the background class).

---

[2]https://github.com/nmndeep/revisiting-at
[3]https://github.com/facebookresearch/ConvNeXt/blob/main/semantic_segmentation/configs/convnext

### B.3 Initialization with Pre-trained Backbones

PIR-AT uses pretrained IMAGENET models as an initialization for the backbone. The robust models are sourced from Singh et al. (2023) (see Table 6). Since, more IMAGENET pre-trained robust models are available on Croce et al. (2021), it does not cost us any additional pre-training cost. In semantic segmentation literature most modern works (Liu et al., 2022; Strudel et al., 2021), use clean IMAGENET pretrained models as initialization for the backbone, making PIR-AT's usage of robust backbone a natural choice. One can further reduce the cost of pre-training by using robust models trained for either 1-step (Debenedetti et al., 2023) or 2-step (Singh et al., 2023) adversarial training, which is the common budget for robust IMAGENET training. For our UPerNet + ConvNeXt-S PIR-AT model for PASCAL-VOC, we also use the 2-step 50 epoch IMAGENET trained model from Singh et al. (2023) as initialization. Using such low-cost pretrained backbones works well, as this model in Table 3 achieves better or similar robust accuracy as the 300 epoch 2-step IMAGENET pretrained ConvNeXt-T in the same table.

Table 7: **APGD consistently outperforms PGD** for two existing attacks, SegPGD (Gu et al., 2022) and CosPGD (Agnihotri and Keuper, 2023), on semantic segmentation. We report ACC and mIOU after the adversarial attack, at various strengths of $\epsilon_\infty$ for our robust UPerNet + ConvNeXt-T model trained for 50 epochs with PIR-AT for PASCAL-VOC at $\epsilon_\infty = 4/255$.

| $\epsilon_\infty$ | $\mathcal{L}_{\text{Bal-CE}}$ | | | | $\mathcal{L}_{\text{CosSim-CE}}$ | | | |
| | SegPGD | | SegAPGD | | CosPGD | | CosAPGD | |
| | ACC | mIOU | ACC | mIOU | ACC | mIOU | ACC | mIOU |
|---|---|---|---|---|---|---|---|---|
| 4/255 | **88.6** | 65.0 | 88.7 | **64.8** | 89.0 | 65.5 | **88.9** | **65.4** |
| 8/255 | 74.6 | **39.4** | **74.2** | 41.3 | 78.2 | 47.5 | **77.8** | **47.3** |
| 12/255 | 45.8 | 15.0 | **43.3** | **14.9** | 58.2 | 28.3 | **56.2** | **26.4** |
| 16/255 | 26.2 | 7.1 | **20.7** | **5.7** | 38.0 | 17.2 | **34.0** | **15.3** |

## C    Additional Experiments

We present additional studies of the properties of our SEA scheme and of the robust models.

### C.1    AGPD vs PGD for attacks on semantic segmentation

For the results in Table 7, the step-sizes are (5e-3, 1e-2, 1e-2, 1e-2) for radii (4/255, 8/255, 12/255, 16/255) resp. and we train for 100 iterations[4].

We show in Table 7 the results of PGD vs APGD (Croce and Hein, 2020) both using 100 iterations for the optimization of the existing loss functions used in SegPGD (Gu et al., 2022) and CosPGD (Agnihotri and Keuper, 2023) for our robust UPerNet + ConvNeXt-T model trained for 50 epochs with PIR-AT for $\epsilon_\infty = 4/255$. APGD consistently improves over PGD for both losses, in particular for robustness evaluation at $\epsilon_\infty = 12/255$ the differences are more than $2\%$ in average robust accuracy and for $\epsilon_\infty = 16/255$ even larger than $4\%$. Thus for a reliable robustness evaluation the use of APGD as optimization scheme is crucial.

### C.2    Analysis of SEA

**Selection of losses.** In Table 2 we have shown the performance of the attacks (APGD with 100 iterations) with six loss functions, and their worst-case. We then selected the best four of those ($\mathcal{L}_{\text{Mask-CE}}, \mathcal{L}_{\text{Bal-CE}}, \mathcal{L}_{\text{JS}}, \mathcal{L}_{\text{Mask-Sph.}}$) to be included in SEA. In Table 8 we additionally compare the robustness computed as worst-case of either all attacks (six runs, one per loss) or the four attacks with objective functions included in SEA. For both clean and robust models, using the two additional losses does not improve performance of the ensemble attack, while increasing run-time significantly.

---

[4]For CosPGD, the authors suggested a step-size of 0.03 at $\epsilon = 8/255$, we found 0.01 yields a stronger attack.

Table 8: **Loss selection for SEA.** We compare taking the worst-case over all six losses from Table 2 APGD with 100 iterations versus using only the subset of the four losses part of SEA (average accuracy and mIOU are shown). The models are UPerNet with ConvNeXt-T backbone, clean trained or adversarially trained with PIR-AT for PASCAL-VOC at $\epsilon_\infty = 4/255$. Leaving out two losses does not substantially impact the performance while considerably reduces the computational cost.

| Subset $\epsilon_\infty \rightarrow$ | 4/255 | | 8/255 | | 12/255 | | 16/255 | |
|---|---|---|---|---|---|---|---|---|
| **model:** clean, 50 epochs | | | | | | | | |
| all losses | 71.1 | 39.9 | 32.5 | 12.1 | 5.3 | 1.2 | 0.1 | 0.0 |
| four losses | 71.2 | 40.0 | 32.6 | 12.1 | 5.3 | 1.2 | 0.1 | 0.0 |
| **model:** PIR-AT 5 step, 50 epochs | | | | | | | | |
| all losses | 88.3 | 64.4 | 72.3 | 38.4 | 31.9 | 8.4 | 6.4 | 1.1 |
| four losses | 88.3 | 64.6 | 72.3 | 38.4 | 31.9 | 9.0 | 6.4 | 1.2 |

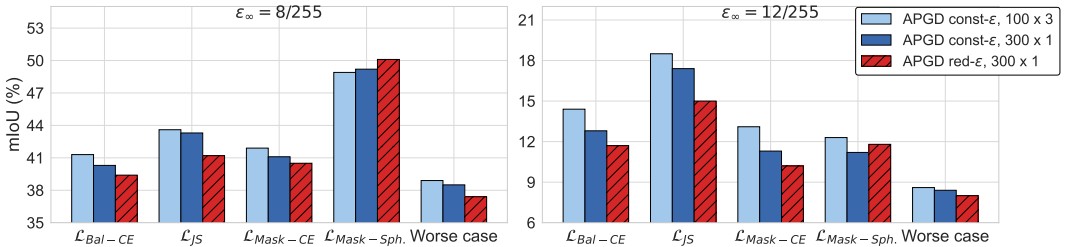

Figure 3: **Comparison of const-$\epsilon$- and red-$\epsilon$ optimization schemes for mIOU.** Balanced attack accuracy for the robust PIR-AT trained UPerNet + ConvNeXt-T model from Table 1 trained on PASCAL-VOC, across different losses for the same iteration budget. The radius reduction (red-$\epsilon$) scheme performs best across all losses, except for $\mathcal{L}_{\text{Mask-Sph.}}$, and $\epsilon_\infty$ and even the worst-case over all losses improves.

Moreover, we show below that each of the remaining four losses positively contributes to the results of SEA, which justifies the choice of including them.

**Effect of reducing the radius.** We complement the comparison of const-$\epsilon$ and red-$\epsilon$ schemes provided in Sec. 2.4 by showing the different robust mIOU achieved by the various algorithms. In Fig. 3 one can observe that, consistently with what reported for average accuracy in Fig. 2, reducing the value of $\epsilon$ (red-$\epsilon$ APGD) outperforms in most cases the other schemes. Moreover, we repeat the same comparison on the clean model on PASCAL-VOC, with $\epsilon = 0.5/255$. Fig. 4 shows that, as for the robust model, the red-$\epsilon$ scheme attains the best results for all losses at exception of the Masked Spherical loss.

**Contribution of individual components in SEA.** To assess how much each loss contributes to the final performance of SEA, we report the individual performance (as average pixel accuracy after attack) at different $\epsilon_\infty$ in Table 9, using robust models on PASCAL-VOC and ADE20K. We recall that each loss is optimized with 300 iterations of red-$\epsilon$ APGD. Additionally, we report the worst-case average pixel accuracy over subgroups of 3 out of 4 losses, and the worst-case of all 4, i.e. SEA. Although subset A works well for both datasets at $\epsilon_\infty \in \{4/255, 8/255\}$, it is not as effective as subsets B and C for higher $\epsilon_\infty$, and vice-versa. Interestingly, $\mathcal{L}_{\text{Mask-Sph.}}$ does not yield the best individual results in any case, but the excluding it from the worst-case computation (subset A) significantly degrades the performance. Hence, we conclude that the four losses show their effectiveness in varied setups, as alluded to in Sec. 2.2, which allows us to have effective attacks across the entire range of $\epsilon_\infty$.

**More iterations.** We also explore the effect of different number of iterations in SEA. In Fig. 5 we show the performance (measured by robust accuracy and mIOU) of SEA with 50, 100, 300 and 500 iterations. There is a substantial improvement going from 50 to 300 iterations in all cases. On further

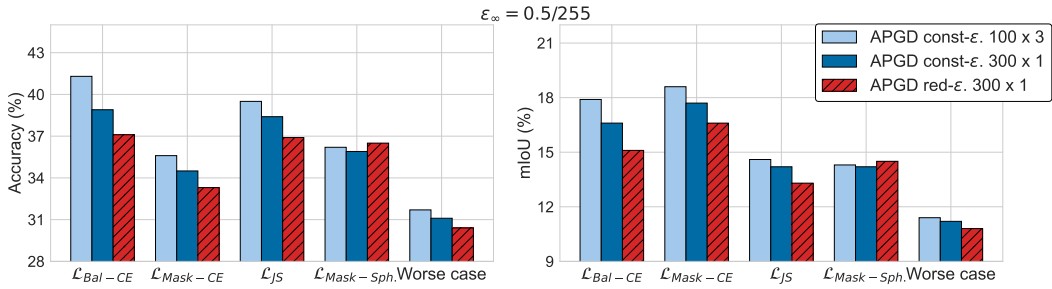

Figure 4: **Comparison of const-$\epsilon$- and red-$\epsilon$ optimization schemes on a clean model.** We repeat the experiment shown in Fig. 2 on the clean model on PASCAL-VOC.

Table 9: **Component analysis for SEA.** We show the individual performance (ACC) of the runs of APGD (red- $\epsilon$) with each loss in SEA for both PASCAL-VOC and ADE20K on 5-step robust models. Additionally we include the worst-case over subsets of 3 out of 4 losses. The best results, among either individual runs or subgroups, are in **bold**.

**A:** $\mathcal{L}_{\text{Mask-CE}} + \mathcal{L}_{\text{Bal-CE}} + \mathcal{L}_{\text{JS}}$  **B:** $\mathcal{L}_{\text{Mask-CE}} + \mathcal{L}_{\text{Bal-CE}} + \mathcal{L}_{\text{Mask-Sph.}}$  **C:** $\mathcal{L}_{\text{Mask-CE}} + \mathcal{L}_{\text{JS}} + \mathcal{L}_{\text{Mask-Sph.}}$

| $\epsilon_\infty$ | individual losses | | | | subsets of three losses | | | all |
|---|---|---|---|---|---|---|---|---|
| | $\mathcal{L}_{\text{Mask-CE}}$ | $\mathcal{L}_{\text{Bal-CE}}$ | $\mathcal{L}_{\text{JS}}$ | $\mathcal{L}_{\text{Mask-Sph.}}$ | A | B | C | (SEA) |
| **model: UPerNet ConvNeXt-T**, PIR-AT, 50 epochs, PASCAL-VOC | | | | | | | | |
| 4/255 | 89.0 | 88.5 | **88.4** | 90.5 | **88.3** | 88.5 | 88.4 | 88.3 |
| 8/255 | 73.7 | **72.9** | 73.8 | 80.7 | **71.6** | 71.7 | 71.8 | 71.2 |
| 12/255 | **31.6** | 35.9 | 38.6 | 38.1 | 29.4 | **27.5** | 27.8 | 27.4 |
| 16/255 | **6.7** | 11.9 | 12.5 | 6.8 | 5.8 | 4.3 | **4.3** | 4.2 |
| **model: UPerNet ConvNeXt-T**, PIR-AT, 128 epochs, ADE20K | | | | | | | | |
| 4/255 | 57.1 | 56.0 | **55.9** | 63.0 | **55.6** | 55.9 | 55.7 | 55.6 |
| 8/255 | **28.6** | **28.6** | 28.7 | 39.6 | **26.5** | 27.1 | 26.7 | 26.4 |
| 12/255 | **4.2** | 4.4 | 4.5 | 4.3 | 3.8 | **3.5** | **3.5** | 3.3 |
| **model: UPerNet ConvNeXt-S**, PIR-AT, 128 epochs, ADE20K | | | | | | | | |
| 4/255 | 58.4 | **57.5** | **57.5** | 63.7 | **57.2** | 57.5 | 57.4 | 57.2 |
| 8/255 | 31.2 | **30.9** | **31.0** | 41.0 | **29.0** | 29.4 | 29.2 | 28.8 |
| 12/255 | **4.7** | 6.3 | 6.3 | 8.0 | 4.3 | 4.1 | **4.0** | 3.9 |
| **model: Segmenter ViT-S**, PIR-AT, 128 epochs, ADE20K | | | | | | | | |
| 4/255 | 56.1 | **54.9** | **54.9** | 60.7 | **54.5** | 54.8 | 54.7 | 54.5 |
| 8/255 | 35.8 | **34.1** | 34.2 | 45.8 | **32.9** | 33.7 | 33.2 | 32.6 |
| 12/255 | **10.4** | 10.9 | 11.2 | 17.5 | **8.2** | 8.8 | 9.1 | 7.6 |

increasing the number of attack iterations to 500, the drop in robust accuracy and mIOU is around 0.1% for both $\ell_\infty$ radii of 8/255 and 12/255. Since going beyond 300 iterations gives no or minimal improvement for significantly higher computational cost, we fix the number of iterations to 300 in SEA.

**Effect of random seed.** We study the impact of the randomness involved in our algorithm (via random starting points for each run) by repeating the evaluation on our robust model on PASCAL-VOC with 5 random seeds. Table 10 shows that the proposed SEA is very stable across all perturbation

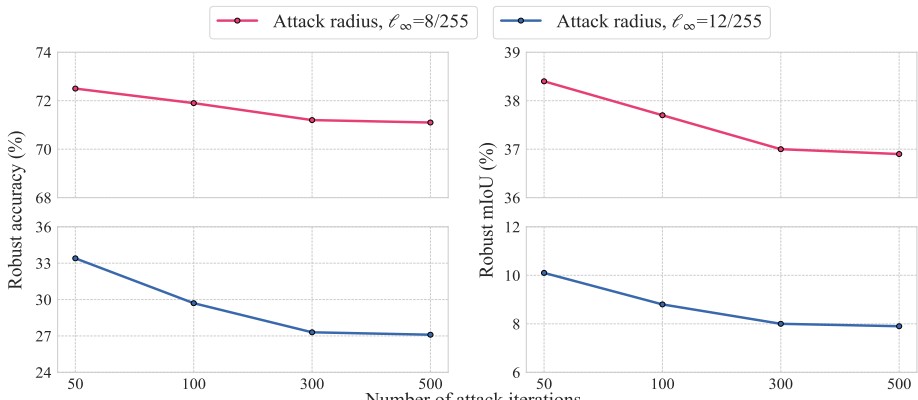

Figure 5: **Influence of number of iterations in SEA.** We show robust average accuracy (left) and mIOU (right) varying the number of iterations in our attack: 300 iterations give the best compute-effectiveness trade-off. We use the 5 step PIR-AT PASCAL-VOC trained ConvNeXt-T backbone UPerNet model.

strengths. It is also interesting to note that all individual losses have negligible variance across the different runs.

### C.3 EXCLUDING THE BACKGROUND CLASS FROM EVALUATION

For ADE20K, we train clean UPerNet + ConvNeXt-T models in two settings, i.e. either ignoring the background class (150 possible classes), which is the standard practice while training clean semantic segmentation models, or to predict it (151 classes). To measure the effect of the additional background class, we can evaluate the performance of both models with only 150 classes (for the one trained on 151 classes, we can exclude the score of the background class when computing the predictions). Training on 150 classes achieves (ACC, mIOU) of (80.4%, 43.8%), compared to (80.2%, 43.8%) for 151. This shows that we do not lose any performance when training with the background class, and the lower clean accuracy of clean trained ADE20K models, (ACC, mIOU) of (75.5%, 41.1%) is due to including the background class when computing the statistics. This also translates to the robust models trained in the 2 step PIR-AT setting. For the robust model, the two settings have (76.6%, 37.8%) and (76.4%, 37.5%) (ACC, mIOU) respectively.

Table 10: **Stability of SEA across different runs.** We report ACC computed on PASCAL-VOC with the 5 step UPerNet model trained with PIR-AT. The mean across 5 runs is shown along with the standard deviation. Across components and perturbation strengths, SEA has a very low variance over random seeds.

**A:** $\mathcal{L}_{\text{Mask-CE}} + \mathcal{L}_{\text{Bal-CE}} + \mathcal{L}_{\text{JS}}$    **B:** $\mathcal{L}_{\text{Mask-CE}} + \mathcal{L}_{\text{Bal-CE}} + \mathcal{L}_{\text{Mask-Sph.}}$    **C:** $\mathcal{L}_{\text{Mask-CE}} + \mathcal{L}_{\text{JS}} + \mathcal{L}_{\text{Mask-Sph.}}$

| $\epsilon_\infty$ | individual losses | | | | subsets of three losses | | | all |
| --- | --- | --- | --- | --- | --- | --- | --- | --- |
| | $\mathcal{L}_{\text{Mask-CE}}$ | $\mathcal{L}_{\text{Bal-CE}}$ | $\mathcal{L}_{\text{JS}}$ | $\mathcal{L}_{\text{Mask-Sph.}}$ | A | B | C | (SEA) |
| **model: UPerNet ConvNeXt-T**, PIR-AT, 50 epochs | | | | | | | | |
| 4/255 | 89.0 ± 0.1 | 88.5 ± 0.1 | **88.4** ± 0.1 | 90.5 ± 0.0 | **88.4** ± 0.1 | 88.5± 0.0 | 88.4 ± 0.0 | 88.3 ± 0.0 |
| 8/255 | 73.6 ± 0.2 | **73.0** ± 0.3 | 73.7± 0.1 | 80.6 ± 0.2 | **71.5**± 0.1 | 71.7± 0.0 | 71.7± 0.1 | 71.2± 0.1 |
| 12/255 | **31.4**± 0.4 | 35.9± 0.2 | 38.2± 0.4 | 38.1± 0.1 | 29.3 ± 0.2 | **27.6**± 0.2 | 27.9± 0.1 | 27.3± 0.1 |
| 16/255 | **6.6**± 0.1 | 11.9± 0.3 | 12.3± 0.2 | 6.9± 0.1 | 5.7± 0.1 | 4.3± 0.1 | **4.3**± 0.1 | 4.2± 0.1 |

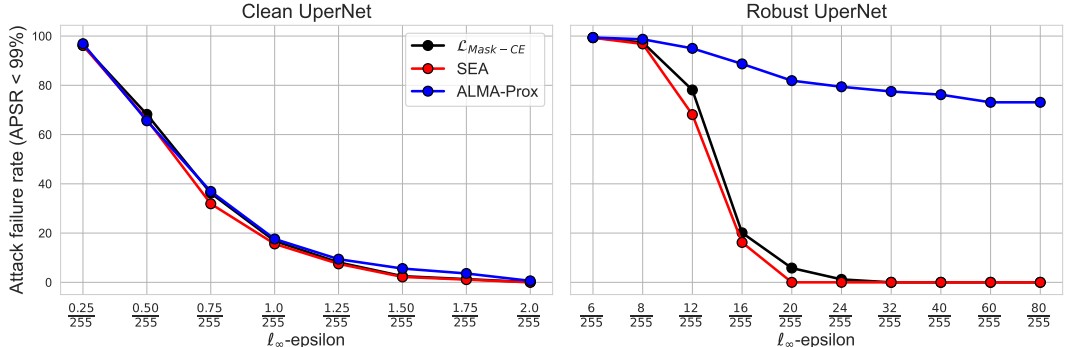

Figure 6: **Comparison to ALMA prox.** We compare APGD with our novel losses ($\mathcal{L}_{\text{Mask-CE}}$) and the ensemble SEA according to the metric used by Rony et al. (2023), which differs from those (ACC and mIOU) we use in the rest of our experiments. In the left plot, the attacks are tested on a clean trained model for the PASCAL-VOC dataset, and in the right plot we test against our robust PIR-AT model.

Table 11: **Tranfer attacks.** We show the effectiveness of various transfer attacks (measured with ACC and mIOU at various radii). For each case we indicate the source and target models. Moreover, we report the evaluation given by white-box attacks as baseline.

| Attack | Source | Target | 0 | | 4/255 | | 8/255 | | 12/255 | |
|---|---|---|---|---|---|---|---|---|---|---|
| **ADE20K, Segmenter with ViT-S backbone** | | | | | | | | | | |
| APGD w/ $\mathcal{L}_{\text{Masked-CE}}$ | clean | PIR-AT | 69.1 | 28.7 | 68.8 | 28.3 | 68.6 | 28.0 | 68.3 | 27.8 |
| APGD w/ $\mathcal{L}_{\text{Masked-CE}}$ | AT | PIR-AT | 69.1 | 28.7 | 66.3 | 26.0 | 63.1 | 23.8 | 57.4 | 19.9 |
| SEA (white-box) | PIR-AT | PIR-AT | 69.1 | 28.7 | 54.5 | 16.1 | 32.8 | 7.1 | 8.6 | 1.8 |

## C.4 ADDITIONAL COMPARISONS TO EXISTING ATTACKS

Rony et al. (2023) have recently proposed ALMA prox as an adversarial attack against semantic segmentation models: its goal is to reach, for each image, a fixed success rate threshold (i.e. a certain percentage of mis-classified pixels, in practice 99% is used) with a perturbation of minimal $\ell_\infty$ norm. Thus, the threat model considered by Rony et al. (2023) is not comparable to ours, which aims at reducing average pixel accuracy as much as possible with perturbations of a limited size.

In order to provide a comparison of our algorithms to ALMA prox, we measure the percentage of images for which the attack cannot make 99% of pixels be misclassified with perturbations of $\ell_\infty$-norm smaller than a threshold $\epsilon$ (i.e. the model is considered robust on such images). In this case, lower values indicate stronger attacks. We show in Fig. 6 the results in such metric, at various $\epsilon$, for ALMA prox (default values, 500 iterations), APGD on the Mask-CE loss (300 iterations) and SEA. We test for 160 random images from the PASCAL-VOC dataset using the clean trained UPerNet with a ConvNeXt-T backbone in the left plot and 5-step adversarially trained version of the same model in the right plot Fig. 6. For the clean model (left plot) the three attacks perform similarly, with a slight advantage of SEA at most radii. However, on the robust model (right plot), both APGD on the Mask-CE loss and SEA significantly outperform ALMA prox: for example, APGD, which uses even less iterations than ALMA prox, attains 0% robustness at $32/255$, compared to 77% of ALMA prox. This shows that, even considering a different threat model, our attacks are effective to estimate adversarial robustness.

## C.5 TRANSFER ATTACKS

To complement the evaluation of the robustness of our PIR-AT models, we further test them with transfer attacks from less robust models. In particular, we run APGD on the Masked-CE loss on Segmenter models obtained with either clean training or AT (5 steps) on ADE20K. We then transfer the found perturbations to our PIR-AT (5 steps, 128 epochs), and report robust accuracy and mIOU

in Table 11, together with the results of the white-box SEA on the same model (from Table 3) as baseline. We observe that the transfer attacks are far from the performance of SEA, which further supports the robustness of the PIR-AT models.

## D    ADDITIONAL FIGURES

**Untargeted attacks.** Fig. 7 shows examples of our untargeted attacks at different radii $\epsilon_\infty$ on the clean model for PASCAL-VOC dataset. In particular, we use 300 iterations of red-$\epsilon$ APGD on the $\mathcal{L}_{\text{Mask-CE}}$ loss. The first column presents the original image with the ground truth segmentation mask, The following columns contain the perturbed images and relative predicted segmentation masks for increasing radii ($\epsilon_\infty = 0$ is equivalent to the unperturbed image): one can observe that the model predictions progressively become farther away from the ground truth values. We additionally report the average pixel accuracy for each image. In Fig. 8, we repeat the same visualization for the most robust 5 step 300 epochs PIR-AT model. Note that we use different values of $\epsilon_\infty$ for the two models, i.e. significantly smaller ones for the clean model, following Table 2. Finally, the same setup is employed on the UPerNet + ConvNeXt-T model trained for ADE20K dataset for the illustrations in Fig. 9 (clean model) and Fig. 10 (5-step robust PIR-AT model), and we have similar observations as for the smaller dataset. Again we use smaller radii for the clean model, since it is significantly less robust than the PIR-AT one.

**Targeted attacks.** In Fig. 1 we show examples of the perturbed images and corresponding predictions resulting from targeted attacks. In this case, we run APGD (red-$\epsilon$ scheme with 300 iterations) on the negative JS divergence between the model predictions and the one-hot encoding of the target class. In this way the algorithm optimizes the adversarial perturbation to have all pixels classified in the target class (e.g. "grass" or "sky" in Fig. 1). We note that other losses like cross-entropy can be adapted to obtain a targeted version of SEA, and we leave the exploration of this aspect of our attacks to future work.

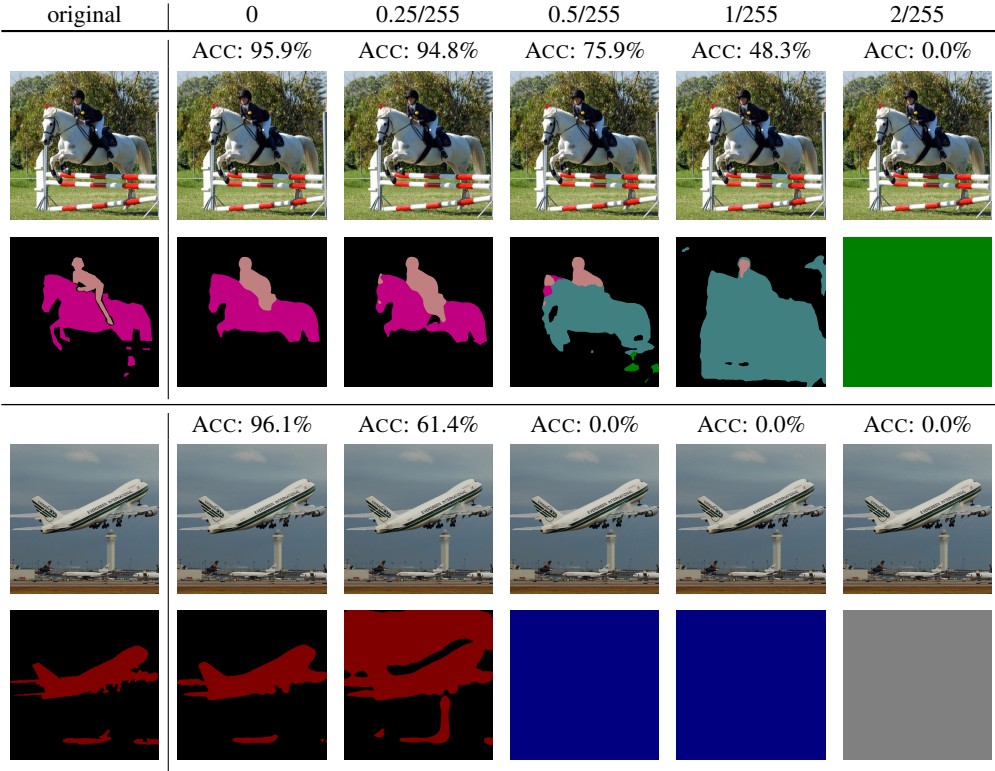

Figure 7: We show the perturbed images, corresponding predicted segmentation masks and average accuracy for increasing radii. The attacks are generated on the clean model on PASCAL-VOC with APGD on $\mathcal{L}_{\text{Mask-CE}}$. We additionally present (first column) the original image and ground truth mask.

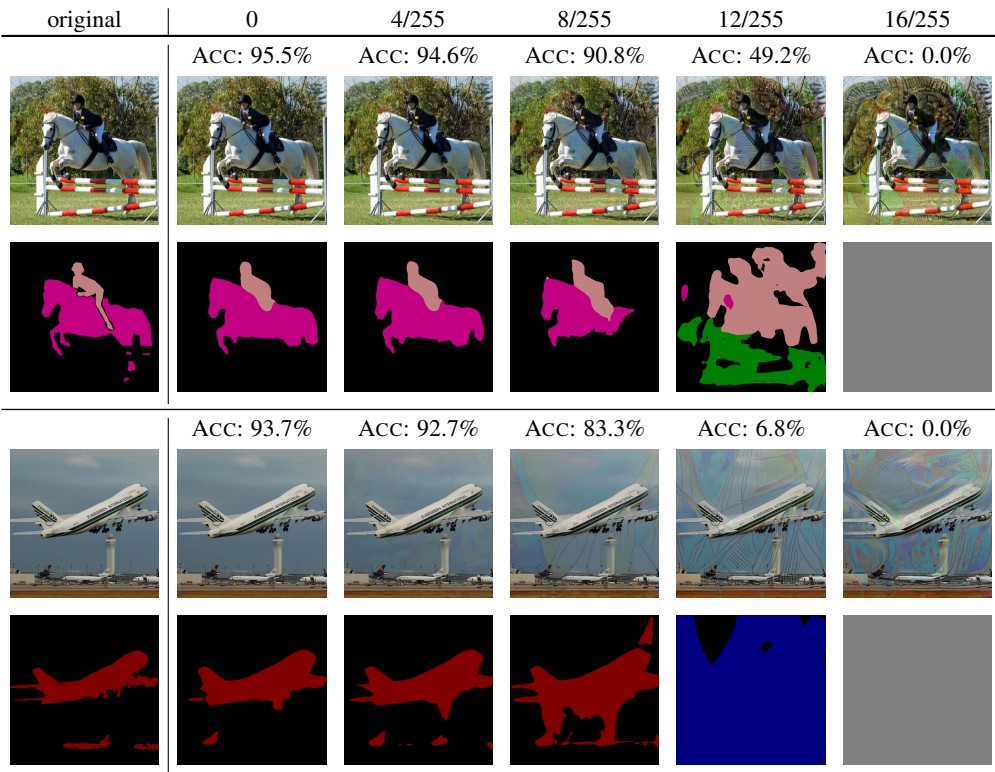

Figure 8: Same setting as in Fig. 7 for the 5-step PIR-AT model. Note the larger radii $\epsilon$ for the attack.

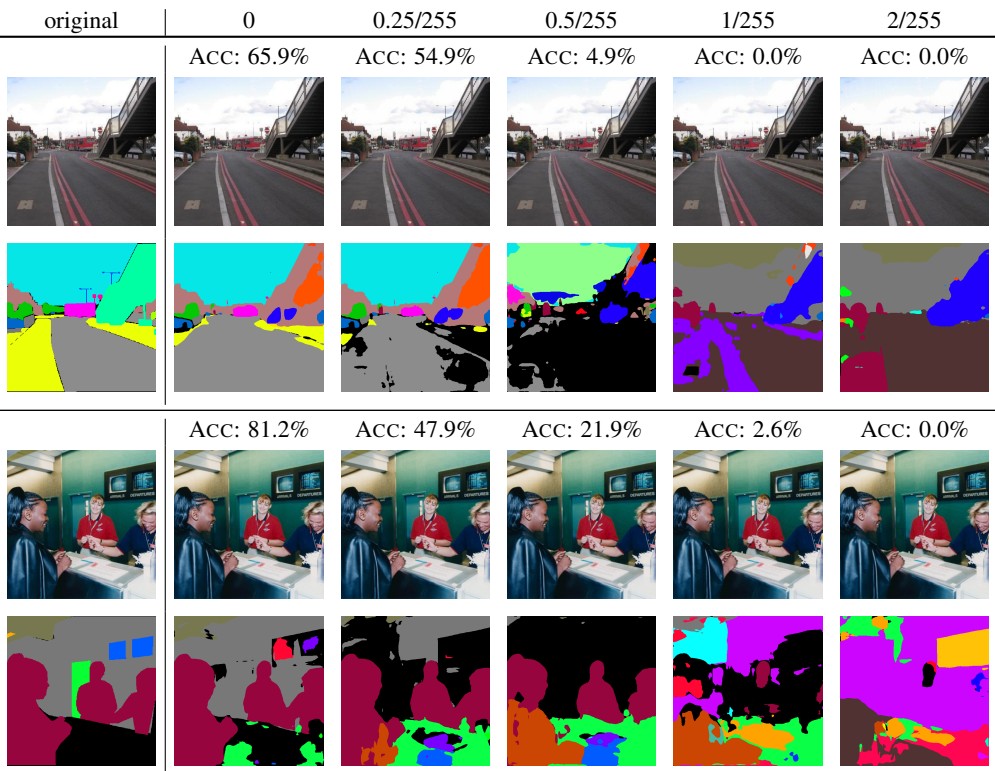

| original | 0 | 0.25/255 | 0.5/255 | 1/255 | 2/255 |
|---|---|---|---|---|---|
| | Acc: 65.9% | Acc: 54.9% | Acc: 4.9% | Acc: 0.0% | Acc: 0.0% |

Figure 9: We show the perturbed images, corresponding predicted segmentation masks and average accuracy for increasing radii. The attacks are generated on the clean model on ADE20K with APGD on $\mathcal{L}_{\text{Mask-CE}}$. We additionally present (first column) the original image and ground truth mask.

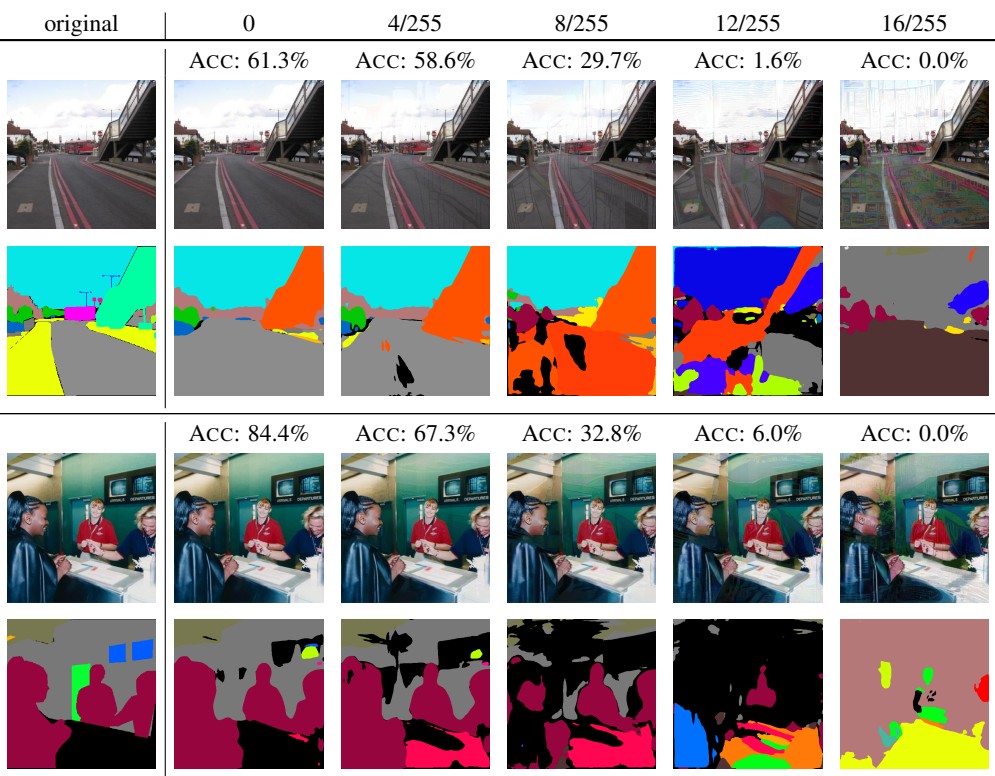

Figure 10: Same setting as in Fig. 9 for the 5 step PIR-AT model. Note the larger radii $\epsilon$ for the attack.

