# OpenReview forum: "Towards Reliable Evaluation and Fast Training of Robust Semantic Segmentation Models"
_ICLR.cc/2024/Conference — Submitted to ICLR 2024_

### Official Review · Reviewer_CxEd · 2023-10-31

**Soundness:** 3 good
**Presentation:** 2 fair
**Contribution:** 2 fair
**Rating:** 5
**Confidence:** 3

**Summary:**

This paper proposes several losses for attacking the semantic segmentation models with adversarial training and an evaluation protocol for benchmarking the adversarial robustness of the segmentation models. This paper also proposes to adopt adversarially pretrained models for segmentation models' initialization. Extensive experiments with various segmentation networks present the effectiveness of the proposed  methods.

**Strengths:**

1. The motivation of this paper for the proposed method is clear and the proposed method is consistent to the motivations.
2. The paper proposes new losses for attacking, evaluating and training semantic segmentation models. The proposed loss and the evaluation protocol could become great baselines for the further works.
3. The proposed method is verified on two popular segmentation networks and two datasets and it presents great insights in this field.

**Weaknesses:**

1. The advantages of the proposed three losses are not well depicted. It would be better if the author could discuss under different scenarios which proposed loss is the best for attacking. Some visual examples would be better.
2. The organization of this paper is confusing. The introduction of some existing works like APGD should be put in the related works. An overview of the method's structure could be added to the beginning of section 2. As section 2 mentions PIR-AT s many times, a short description about PIR-AT model is also necessary.
3. As many related works and ablation studies are mixed in the method part, it is difficult to distinguish the contributions of this work from previous works.
4. The SEA is not presented clearly. Why four losses perform worse than all six losses is not analyzed. How the current four losses are selected is not mentioned. And SEA doesn't discuss how to balance different losses.
5. Previous works[1] have proven that using a better robust initialization model could improve the task model's robustness. How PIR-AT is different from the existing practices is not well presented.

[1]Tianlong Chen, Sijia Liu, Shiyu Chang, Yu Cheng, Lisa Amini, and Zhangyang Wang. Adversarial Robustness: From Self-Supervised Pre-Training to Fine-Tuning. CVPR 2020

**Questions:**

1. Will AT obtain the same performance as PIR-AT if sufficient training time is given?
2. PIR-AT suggests using $L_{\infty}$-robust ImageNet model for initialization. How much computational resources are required to train this model compared to the normal ImageNet model with the same parameters?
3. How the image-wise worst case over all losses in Table 2 is calculated? A short description is expected.
4. Does the conclusion in Figure 2 also generalize to clean models?
5. What is the result of AT with 32 epoch in Figure 5?

---

> ### Author Response · Authors · 2023-11-18
> **Reply to Reviewer CxEd (part 1)**
>
> We thank the Reviewer for the positive and detailed feedback. In the following we address the weaknesses and questions raised in the review.
>
> &nbsp;
>
> **Q: The advantages of the proposed three losses are not well depicted. It would be better if the author could discuss under different scenarios which proposed loss is the best for attacking. Some visual examples would be better.**
>
> The (theoretical) discussion of benefits and weaknesses for each loss in Sec. 2.2 suggests that one main difference among losses is how they balance the weight of different pixels in the objective function. On one extreme, the plain cross-entropy maximizes the loss for all pixels independently of whether they are misclassified, and assign them the same importance. Conversely, the masked losses exclude (via the mask) the misclassified pixels from the objective function, with the danger of reverting back the successful perturbations. As middle ground, losses like the JS divergence assign a weight to each pixel based on how “confidently” they are misclassified. We conjecture that for radii where robustness is low, masked losses help focusing on the remaining pixels, and already misclassified pixels are hardly reverted since they are far from the decision boundary. Conversely, at smaller radii achieving confident misclassification is harder (since the perturbations are smaller), and most pixels are still correctly classified or misclassified but close to the decision boundary: then it becomes more important to balance all of them in the loss, hence losses like JS divergence are more effective. This hypothesis is in line with the empirical results in Table 2. We are happy to add this more detailed discussion to the paper.
>
> Finally, if the Reviewer could clarify what they mean for “visual examples”, we will address that point directly.
>
> &nbsp;
>
> **Q: The organization of this paper is confusing. The introduction of some existing works like APGD should be put in the related works. An overview of the method's structure could be added to the beginning of section 2. As section 2 mentions PIR-AT s many times, a short description about PIR-AT model is also necessary.**\
> **Q: As many related works and ablation studies are mixed in the method part, it is difficult to distinguish the contributions of this work from previous works.**
>
> Thanks for the suggestions about the presentation. We agree with the Reviewer that the definition of PIR-AT comes late in the paper, we will add a short description earlier on in the revision. We will also add more detail of SEA at the start of Sec. 2, and discuss APGD in the related works.
>
> &nbsp;
>
> **Q: The SEA is not presented clearly. Why four losses perform worse than all six losses is not analyzed. How the current four losses are selected is not mentioned. And SEA doesn't discuss how to balance different losses.**
>
> We introduce the details of SEA in Sec. 2.4, combining the loss functions analyzed in Sec. 2.2 and the optimization algorithm from Sec. 2.3. SEA is based on a worst-case evaluation: this means that, for each image, the strongest attack among those obtained running APGD on the different losses is considered. Therefore, having more losses can only improve its performance (since it has a strictly larger set of attacks to choose among). However, each additional loss requires additional computational cost. Then, as mentioned in Sec. 2.4, we select 4 losses which allow our ensemble SEA to perform already on par with using all 6 losses, while saving ⅓ of runtime. This is further illustrated in one of the ablation studies in App. C.2, which shows that the improvement given by 6 over 4 losses is minimal (<0.1% in robust accuracy). Finally, since the losses are used for independent runs, it is not necessary to balance them.
>
> &nbsp;
>
> **Previous works[1] have proven that using a better robust initialization model could improve the task model's robustness. How PIR-AT is different from the existing practices is not well presented.**
>
> [1] studies the effect of self-supervised pre-training for robustness of image classifiers: they show that adversarial pre-training has no significant advantage over clean pre-training to provide best robust accuracy on CIFAR-10 when full adversarial fine-tuning is used (i.e. all network parameters are updated with adversarial training), as illustrated by Table 3 in [1]. In our case, we show that leveraging pre-trained image classifiers which are publicly available, e.g. in RobustBench, we can largely improve the performance of adversarial training on semantic segmentation tasks. This means that in this scenario, unlike in [1], using a robust backbone has a significant impact on the resulting robustness in the target task.

---

> > ### Author Response · Authors · 2023-11-18
> > **Reply to Reviewer CxEd (part 2)**
> >
> > **Q: Will AT obtain the same performance as PIR-AT if sufficient training time is given?**
> >
> > The ablation study in Table 4 shows that, on Pascal-VOC, AT can reach the same level of robustness obtained by PIR-AT with 6x longer training (300 vs 50 epochs). However, on the more challenging ADE-20k, AT with 128 epochs cannot match the robustness of PIR-AT with 32 epochs, for both UPerNet and Segmenter architectures. This emphasizes the importance of using a robust backbone for initialization.
> >
> > &nbsp;
> >
> > **Q: PIR-AT suggests using Linf-robust ImageNet model for initialization. How much computational resources are required to train this model compared to the normal ImageNet model with the same parameters?**
> >
> > Adversarial training on ImageNet is typically carried out with 1 or 2 steps of PGD, which means 2-3x higher computational cost than clean training for the same number of epochs (Singh et al., 2023). However, we emphasize that we did **not** train the robust ImageNet classifiers, but took some of those open sourced by prior works. In fact, we want to stress that nowadays the cost of the initialization is virtually zero, since plenty of both clean and robust ImageNet models are publicly available. For example, there are already 18 robust models for ImageNet in [RobustBench](https://robustbench.github.io/#div_imagenet_Linf_heading) with various sizes and architectures.
> >
> > &nbsp;
> >
> > **Q: How the image-wise worst case over all losses in Table 2 is calculated? A short description is expected.**
> >
> > Please refer to the paragraph “Metrics” in Sec. 2.1: for the worst-case over multiple attacks, we select the perturbation that gives the lowest average accuracy for each image and use it to compute the mIoU.
> >
> > &nbsp;
> >
> > **Q: Does the conclusion in Figure 2 also generalize to clean models?**
> >
> > We repeated the ablation study illustrated in Fig. 2 on the clean model ($\epsilon=0.5/255$ is used), and we observe results consistent with those obtained for the robust model. In particular, the red-$\epsilon$ scheme yields the lowest (best) robust accuracy for both the individual losses, except the Masked-Spherical loss, and worst-case evaluation. We will add this comparison in the revision of the paper.
> >
> > &nbsp;
> >
> > **Q: What is the result of AT with 32 epoch in Figure 5?**
> >
> > Fig. 5 includes models on Pascal-VOC, which are trained with either 50 or 300 epochs as reported in Table 4 (in particular, for Fig. 5 we use the clean model and the PIR-AT model trained with 5 steps for 300 epochs). Conversely, we use 32 epochs on ADE-20k (Table 4). Could the Reviewer please clarify which is the suggested experiment?

---

> > > ### Author Response · Authors · 2023-11-23
> > > **Discussion period ending soon**
> > >
> > > Dear Reviewer CxEd,
> > >
> > > Since the discussion period is ending soon, we would be happy to know whether the rebuttal and the additional experiments we provided in the revised paper have fully addressed your concerns, and are worth adjusting your score.
> > >
> > > Thanks for your work!

---

> ### Comment · Reviewer_CxEd · 2023-11-23
>
> I would highly appreciate the efforts the authors have put for rebuttal. Most of my concerns are addressed. I appreciate the workload of this paper for developing segmentation model evaluation protocol and the extensive experiments with various losses. However, as the proposed losses are all from existing works, SEA seems to be running the models with various losses combinations, and PIR-AT is just a common practice to apply an adversarially pre-trained model, I agree with the point of Reviewer pqj9 that the current presentation would fit for a workshop paper. Maybe after improving the writing this paper could be accepted by other top-level conferences, I believe the current format is not suitable for acceptance for ICLR. I would modify my rating to 5 to illustrate my point.
>
> BTW, after reading the rebuttal and other reviewers' comments, I raise more concerns about SEA. Table 8 is only verified on Pascal-VOC with one model, which somehow harm the reliability of the claim that four losses perform on par with all six losses for a generalized setting. Table 9 also indicates that on different dataset and models, different loss combinations may result in different conclusion(subset C performs best on UPerNet while subset A performs best on Segmentor for 12/255).
>
> For W1's *visual examples* I mean the visual images after the perturbation.
>
> For Q5's *result of AT with 32 epoch*, I made a mistake and I consider it would be better to add results of AT with 32 epoch for ADE20K, UPerNet with ConvNeXt-T backbone in Table 4 so that it would be more complete.

---

### Official Review · Reviewer_tMpk · 2023-11-01

**Soundness:** 3 good
**Presentation:** 3 good
**Contribution:** 3 good
**Rating:** 6
**Confidence:** 3

**Summary:**

This paper studies the l_{\limit} white-box adversarial attacks for semantic segmentation model. By discussing the loss functions used in semantic segmentation, i.e., pixel-level cross entropy loss, this paper shows the difficulty of adversarial attacks for semantic segmentation, than image classification model. Besides, this paper also proposes and compares 4 loss functions for semantic segmentation. As a result, to achieve higher attack performance, the proposed method combines 4 loss functions as SEA attack. Finally, this paper also studies the defense techniques for the above attacks. Comparing the proposed method with SegPGD and CosPGD, this work shows stronger attack performance won ADE20K and Pascal VOC. Finally, this paper also presents the comparison between the proposed defense PIR-AT with AT on several network architectures.

**Strengths:**

+ The overall work is solid, that the proposed method starts from the analysis of loss functions for semantic segmentation. Besides, 4 different loss functions are compared, and then Semantic Ensemble Attack (SEA) is proposed, which is interesting.

+ The evaluation is conducted under different attack strengths and network architectures.

+ Different optimization methods are discussed for adversarial attacks, that fewer computation costs are needed.

**Weaknesses:**

- This paper needs to discuss more about PIR-AT. It is only mentioned that this paper proposes Pre-trained ImageNet Robust Models. What is this method in detail?

- What is the motivation to limit this paper to focus on l_{\limit} threat model?

**Questions:**

See weakness section.

---

> ### Author Response · Authors · 2023-11-18
> **Reply to Reviewer tMpk**
>
> We thank the Reviewer for the positive feedback. In the following we address the weaknesses pointed out in the review.
>
> &nbsp;
>
> **Q: This paper needs to discuss more about PIR-AT. It is only mentioned that this paper proposes Pre-trained ImageNet Robust Models. What is this method in detail?**
>
> With PIR-AT (as described in Sec. 3.1) we propose to use a robust image classifier, i.e. obtained with adversarial training, on ImageNet as initialization of the backbone of the segmentation model. Such initialization allows adversarial training on the segmentation task to achieve higher robustness than existing methods, even with lower computational cost. As shown in Table 3 and Table 4, our PIR-AT models outperform DDC-AT (Xu et al., 2021), SegPGD-AT (Gu et al., 2022), and match or improve over AT from clean initialization with 4-6x fewer training epochs.
>
> &nbsp;
>
> **Q: What is the motivation to limit this paper to focus on l_{\limit} threat model?**
>
> We focus on the $\ell_\infty$-threat model since this is the most popular in the literature: in fact, for image classification, the large majority of defenses reported in [RobustBench](https://robustbench.github.io/#div_imagenet_Linf_heading) are for $\ell_\infty$, and all of those on ImageNet (which we use as initialization of the model backbone in PIR-AT). Similarly, the closest prior work for robust semantic segmentation, SegPGD (Gu et al., 2022), focuses on $\ell_\infty$-bounded attacks. However, we think that SEA might be easily extended to other $\ell_p$-threat models since APGD provides versions for such cases, which could be an interesting direction to explore for future work.

---

> > ### Author Response · Authors · 2023-11-23
> > **Discussion period ending soon**
> >
> > Dear Reviewer tMpk,
> >
> > Since the discussion period is ending soon, we would be happy to know whether the rebuttal and revised paper have fully addressed your concerns, or if you have any additional question.
> >
> > Thanks for your work!

---

### Official Review · Reviewer_pqj9 · 2023-11-01

**Soundness:** 2 fair
**Presentation:** 3 good
**Contribution:** 2 fair
**Rating:** 5
**Confidence:** 4

**Summary:**

The paper focus on making robust evaluation of semantic segmentation models against L-inf adversarial attacks and show a straightforward approach to train robust segmentation models faster. Authors review existing measures like Jensen-Shannnon divergence, Masked cross-entropy and Masked spherical loss for their applicability as adversarial objectives for semantic segmentation task. They show that these objectives serve as better robustness evaluators than previously utilized objectives in the literature. They make three optimization related decisions: 1) replacing PGD with APGD, 2) progressively reduce attack radius and, 3) train for more iterations. Finally, they propose Segmentation Ensemble Attack (SEA) to evaluate models with different losses utilizing APGD and optimize for more iterations. Furthermore, to improve speed and efficiency of adversarial training, they initialize backbone of semantic segmentation models with ImageNet Robust Models and show a significant improvement on the adversarial robustness.

**Strengths:**

-	Paper is well written and easy to understand.

-	Motivation is clearly delivered.

-	Discussed and reviewed different measures for their suitability for adversarial loss.

-	Results show that the candidate losses mentioned in the paper attack the model better.

-	Backbones initialized with ImageNet Robust Models provide higher adversarial training robust accuracy than using standard ImageNet model.

**Weaknesses:**

My major concern is the lack of originality and novelty in the paper. All the losses, optimizations tricks, and robust models utilized in the paper are obtained from the existing literature (authors have cited the prior works sufficiently). There are no novel methodological contributions presented in the paper. Paper detail as different existing components collectively utilized to obtain better results. The findings shown in the paper are interesting. However, I believe that they alone do not support to meet the standards for accepting at a conference. Novel methodological contribution regarding losses or obtained robust models would be appreciated.

**Questions:**

My concern mainly targets the core essence of the paper i.e. lack of originality and technical novelty. I appreciate the authors for conducting this study. I believe this paper would fit for a workshop submission.

---

> ### Author Response · Authors · 2023-11-18
> **Reply to Reviewer pqj9**
>
> We thank the Reviewer for the feedback. Below we address their concern about novelty raised in the review.
>
> &nbsp;
>
> **Q: My major concern is the lack of originality and novelty in the paper. All the losses, optimizations tricks, and robust models utilized in the paper are obtained from the existing literature (authors have cited the prior works sufficiently). There are no novel methodological contributions presented in the paper. Paper detail as different existing components collectively utilized to obtain better results. The findings shown in the paper are interesting. However, I believe that they alone do not support to meet the standards for accepting at a conference. Novel methodological contribution regarding losses or obtained robust models would be appreciated.**
>
> We appreciate that the Reviewer finds our results interesting and our paper well written. However, we think that the contributions of this paper are more than sufficient - in fact this paper sets completely new standards regarding robust semantic segmentation:
>
> 1. on the level of robustness evaluation, our attack ensemble SEA shows that even the previously best attack SegPGD overestimates robustness by more than 17% (Table 1, PIR-AT model, evaluation at 12/255). As already observed with AutoAttack, a good and reliable attack for robustness evaluation enables better comparisons of new defense techniques and thus faster progress in the field.
>
> 2. we show how to train robust semantic segmentation models and improve by more than 20 points in mIoU compared to the SegPGD-AT paper (their reported numbers are even for a weaker attack than ours). In the previous paper DDC-AT (Xu et al., 2021), a sophisticated method was presented, which unfortunately proved to be completely unrobust. Therefore, we believe that it is an important contribution of this work to show how to train robust semantic segmentation models using pre-trained robust ImageNet backbones, and even save computational time. Also, this shows that seemingly strong methodological contributions can end up being completely useless.
>
> 3. we make all our code and models available (in contrast to SegPGD-AT) to foster research in this area as well as reproducibility.
>
> Additionally to these points, we have clear methodological contributions as we analyze why the plain cross-entropy loss, one of the standard losses for attacks for image classification, does not work well for semantic segmentation. At the same time we show that the Jensen-Shannon divergence, a loss not used for adversarial attacks before, has exactly the properties needed for semantic segmentation (the gradient vanishes as the pixel is maximally misclassified) and thus does not require any masking. Thus while the losses themselves are not novel, there is a clear novelty in our analysis and in the justification why these losses are good for semantic segmentation.

---

> > ### Comment · Reviewer_pqj9 · 2023-11-23
> >
> > I have read all the reviews and authors response. I understand that this work show overestimation of robustness by prior works and suggest a robust ImageNet backbone for training. I agree on authors point regarding usage of Jensen-Shannon divergence for semantic segmentation. However, as I find the results interesting, but they are obtained from collective utilization of prior methods (as in conducting ablation study of different methods) without additional novel insights and contributions. After reconsidering authors response for all the reviewers, I increase my rating to 5 but not confident to consider it to be a potential contribution to the conference.

---

> > > ### Author Response · Authors · 2023-11-23
> > >
> > > We thank the Reviewer for taking the time of going through all reviews and the rebuttal, and updating their score.
> > >
> > > We are glad that the Reviewer acknowledges the relevance of the results in our work about outperforming existing attacks and developing SOTA robust segmentation models, as well as the novelty of the JS loss. However, we respectfully disagree on the overall judgement of our paper: we think we provide both novel insights, significant empirical results and methodological contributions, which will be of interest and helpful to the community.
> > >
> > > Thanks for your work!

---

### Official Review · Reviewer_hjSQ · 2023-11-01

**Soundness:** 2 fair
**Presentation:** 2 fair
**Contribution:** 2 fair
**Rating:** 5
**Confidence:** 4

**Summary:**

In this paper, the authors propose an ensemble of adversarial attacks,  like in the AutoAttack framework, containing attacks with different loss functions for the task of semantic segmentation. In particular, they empirically show that existing loss functions for the task of semantic segmentation overestimate the confidence of robust models. Furthermore, they also train a robust model by utilizing the robust backbones from image classification literature, which significantly boosts performance while saving computing power

**Strengths:**

- The presentation of the paper is good
- The empirical boost in performance is consistent across the board for different models
- It is interesting to see the benefit that comes with the robust initialization using pre-trained Imagnet models

**Weaknesses:**

- My major concern is the limited novelty, as the explored loss functions are not new. Although JS divergence, masked CE loss, and masked spherical loss have not been commonly used in the context of segmentation attacks, in my view, this appears to be a simple 'plug and play' of loss functions


- The conducted attacks are white-box, and the absence of black-box evaluation is a significant limitation

- The paper only considers untargeted attacks, and it would be useful to extend the analysis to targeted attacks to showcase the strength of the proposed attack method.


- The authors could conduct experiments to evaluate the transferability of the proposed attack to other models and compare it against the baseline PGD/CosPGD/SegPGD attacks.

**Questions:**

- Please see my comments in the Weakness section.

- Why is AT in Section 3 performed with the PGD attack baseline?  It would be interesting to use stronger attacks during the AT to develop even stronger robust models

- How did you choose the budget scheme of 3:3:4 in the progressive reduction of epsilon approach?

---

> ### Author Response · Authors · 2023-11-18
> **Reply to Reviewer hjSQ (part 1)**
>
> We thank the Reviewer for the detailed feedback. In the following we address the weaknesses and questions raised in the review.
> &nbsp;
>
> **Q: My major concern is the limited novelty, as the explored loss functions are not new. Although JS divergence, masked CE loss, and masked spherical loss have not been commonly used in the context of segmentation attacks, in my view, this appears to be a simple 'plug and play' of loss functions**
>
> In the discussion around the usage of different losses, we think we have clear methodological contributions: in fact, we analyze why the standard cross-entropy loss, one of the standard losses for attacks for image classification, does not work well for semantic segmentation. At the same time we show that the Jensen-Shannon divergence, a loss not used for adversarial attacks before, has instead the properties needed for semantic segmentation (the gradient vanishes as the pixel is maximally misclassified) and then does not require any masking. Thus, while the losses themselves are not novel, there is a clear novelty in our analysis and in the justification why these losses are good for semantic segmentation.
>
> Moreover, we uncover the complementarity of different losses, which is key, together with the improvements in the optimization algorithm, for our ensemble attack SEA to outperform the existing SOTA attacks by more than 17% (Table 1, PIR-AT model, evaluation at 12/255).
>
> &nbsp;
>
> **Q: The conducted attacks are white-box, and the absence of black-box evaluation is a significant limitation**
>
> To extend the evaluation to black-box methods, we tried to adapt Square Attack [(Andriushchenko et al., 2020)](https://arxiv.org/abs/1912.00049), a SOTA score-based black-box attack for image classification, by using its random search-based algorithm to optimize our JS loss. However, the resulting attack has poor performance even on clean models. We hypothesize that the localized updates in Square Attack are not sufficiently effective to optimize pixelwise losses. Similarly we tested using gradient estimation via finite difference [(Ilyas et al., 2018)](https://arxiv.org/abs/1804.08598), but could not get performance close to that of white-box methods. We showed in the paper that balancing gradients of different pixels is already difficult with exact gradients, then we argue that having only approximated gradients is expected to make optimization even more challenging. This shows that finding an effective black-box attack for semantic segmentation requires designing task-specific algorithms, which could be the topic of an independent paper.
>
> As additional black-box evaluation, we tested transfer attacks from less robust models to the PIR-AT. In particular, we run APGD on the Masked-CE loss on Segmenter models obtained with either clean training or AT (5 steps) on ADE-20k. We then transfer the found perturbations to our PIR-AT (5 steps, 128 epochs), and report robust accuracy and mIoU in Table A, together with the results of the white-box SEA on the same model (from Table 3 of the paper) as baseline. We observe that the transfer attacks are far from the performance of SEA, which further supports the robustness of the PIR-AT models. We are happy to add this evaluation (potentially expanded to include more models) in the revision of the paper.
>
> **Table A:** Robust average accuracy and mIoU given by transfer attacks on ADE-20k dataset. In boldface the white-box baseline by SEA.
>
> |source model | target model     | 0    |        | 4/255 |        | 8/255 |       | 12/255 |       |
> |:--------------|:-----------|-----------:|--------:|-------:|--------:|-------:|-------:|--------:|-------:|
> | | |aAcc | (mIou)| aAcc | (mIou)| aAcc | (mIou)|aAcc | (mIoU)|
> | Segmenter clean | Segmenter PIR-AT | 69.1 | (28.7) | 68.8 | (28.3) | 68.6 | (28.0)| 68.3 | (27.8) |
> | Segmenter AT | Segmenter PIR-AT | 69.1 | (28.7) | 66.3 | (26.0) | 63.1 | (23.8) |  57.4 | (19.9) |
> | **Segmenter PIR-AT** | **Segmenter PIR-AT** | 69.1 | (28.7) | 54.5 | (16.1) | 32.8 | (7.1) | 8.6 | (1.8) |

---

> > ### Author Response · Authors · 2023-11-18
> > **Reply to Reviewer hjSQ (part 2)**
> >
> > **Q: The paper only considers untargeted attacks, and it would be useful to extend the analysis to targeted attacks to showcase the strength of the proposed attack method.**
> >
> > We mainly focus on untargeted attacks since these are the most popular threat model in the literature. However, our attacks can be readily extended to the targeted scenario by using the targeted versions of the various loss functions (e.g. one can minimize the cross-entropy loss of each pixel to a class). For example, we already used a targeted version of APGD on the JS divergence loss in Fig. 1, to have the entire image being classified as either “sky” or “grass” (this is also described in App. D). We would be happy to add more such examples in the revision of the paper, and, if possible before the end of the rebuttal, a quantitative evaluation which the Reviewer would consider relevant.
> >
> > Finally, we note that the targeted attacks are a more restrictive threat model than untargeted attacks: this implies that models which are robust to untargeted attacks are also robust to targeted ones with the same perturbation radius.
> >
> > &nbsp;
> >
> > **Q: The authors could conduct experiments to evaluate the transferability of the proposed attack to other models and compare it against the baseline PGD/CosPGD/SegPGD attacks.**
> >
> > As a preliminary experiment we compare the transferability of APGD on the Masked-CE loss and SegPGD on a few models on ADE-20k, and report the results in Table B. SegPGD is slightly better than APGD, especially at larger radii. We hypothesize that this is due to the fact that using a mask stops the optimization for misclassified pixels, but higher confidence misclassifications are expected to transfer better. In this regard, it might be interesting to explore how our different losses influence the transferability of the perturbations. However, we emphasize that transferability is **not** one of the goals of our attacks. In fact, the most effective transfer attacks typically rely on specific techniques to enhance transferability, e.g. [Wang \& He (2021)](https://arxiv.org/abs/2103.15571), [Wang et al. (2021)](https://arxiv.org/abs/2102.00436), [Zhang et al. (2023)](https://arxiv.org/abs/2303.15735), and do not necessarily correspond to the most successful white-box methods.
> >
> > **Table B:** transferability of different attacks on ADE-20k.
> >
> > |attack |source model | target model     | 4/255 |        | 8/255 |       | 12/255 |       |
> > |:----|:--------------|:-----------|-------:|--------:|-------:|-------:|--------:|-------:|
> > | | | |aAcc | (mIou)| aAcc | (mIou)| aAcc | (mIou)|aAcc | (mIoU)|
> > | SegPGD |Segmenter AT | Segmenter PIR-AT | 65.0 | (25.1) | 59.1 | (20.8)| 49.9 |(16.1) |
> > | APGD w/ Masked-CE |Segmenter AT | Segmenter PIR-AT | 69.1 | (28.7) | 66.3 | (26.0) | 63.1 | (23.8) |  57.4 | (19.9) |
> > | SegPGD |Segmenter AT | UPerNet PIR-AT | 67.7 | (28.7) | 63.4 |(25.2) | 55.6| (20.3) |
> > | APGD w/ Masked-CE |Segmenter AT | UPerNet PIR-AT | 68.6| (30.0) | 64.4 |(26.4) | 60.9 |(23.7) |
> >
> > &nbsp;
> >
> > **Q: Why is AT in Section 3 performed with the PGD attack baseline? It would be interesting to use stronger attacks during the AT to develop even stronger robust models**
> >
> > We believe that it is an important contribution of this work to show how it is possible to train robust semantic segmentation models with standard PGD-based AT. As a result, our PIR-AT models significantly outperform previous, more sophisticated, methods such as SegPGD-AT and DDC-AT (see Table 3).
> >
> > We agree with the Reviewer that future work might focus on optimizing the attack in AT. In fact, we briefly tested the effect of using alternative losses for PGD in AT, but this did not provide benefits (see the experimental setup paragraph at the beginning of Sec. 3): we hypothesize that, when using attacks with only a few iterations (2-5) as in adversarial training, optimizing the loss for all pixels without re-weighting or masking, as in the plain cross-entropy loss, is anyway the better choice.
> >
> > &nbsp;
> >
> > **Q: How did you choose the budget scheme of 3:3:4 in the progressive reduction of epsilon approach?**
> >
> > We followed Croce & Hein (2021) for this split. Moreover, this scheme gives sufficient iterations for the attack to be effective at each radius, with a slight bias towards the final and target radius.

---

> > > ### Author Response · Authors · 2023-11-23
> > > **Discussion period ending soon**
> > >
> > > Dear Reviewer hjSQ,
> > >
> > > Since the discussion period is ending soon, we would be happy to know whether the rebuttal and the additional experiments we provided have fully addressed your concerns, and are worth reconsidering your score.
> > >
> > > Thanks for your work!

---

### Author Response · Authors · 2023-11-18
**General response to Reviewers**

We thank again all Reviewers for their comments.

We have provided individual responses to the questions raised in the reviews. We are happy to address additional comments from the Reviewers before the end of the rebuttal phase. Moreover, we will upload a revised version of the paper including the experiments and updates mentioned in the rebuttal in the next days.

---

### Author Response · Authors · 2023-11-22
**Revision and reminder**

Dear Reviewers,

we have now uploaded the revised paper which integrates all the comments and additional results mentioned in the rebuttal (highlighted in orange).

Since the discussion period is closing soon, we would like to know if the rebuttal has addressed all concerns expressed in the reviews, and invite the Reviewers to update their scores accordingly. We are happy to discuss in the remaining time any follow-up questions.

Thanks for your work!

---

### Meta-Review · Area_Chair_KtKN · 2023-12-08

**Metareview:**

This paper studies adversarial robustness for semantic segmentation models, and propose efficient training strategies for advesarially robust models, including the use of pre-trained adversarially robust imageNet classifier networks. The reviewers appreciate the experimental results presented in the paper, but have consistent reservations regarding the technical contributions of the paper. The author rebuttal addressed some but not all of the points raised in the reviews, and in particular does not alleviate the conern regarding novelty.

**Justification For Why Not Higher Score:**

The reviewers consistently feel that the paper is lacking enough novel technical contributions, despite having interesting experimental results.  The only positive recommendation is a 6, which does not argue strongly in favor of the paper, and did not respond to the rebuttal.

**Justification For Why Not Lower Score:**

N/A

---

### Decision · Program_Chairs · 2024-01-16

Reject